



# Inversion of multi-angular polarimetric measurements from the ACEPOL campaign: an application of improving aerosol property and hyperspectral ocean color retrievals

Meng Gao[1], Peng-Wang Zhai[2], Bryan A. Franz[3], Kirk Knobelspiesse[3], Amir Ibrahim[1], Brian Cairns[5], Susanne E. Craig[3], Guangliang Fu[6], Otto Hasekamp[6], Yongxiang Hu[4], and P. Jeremy Werdell[3]

[1]SSAI, NASA Goddard Space Flight Center, Code 616, Greenbelt, Maryland 20771, USA
[2]JCET/Physics Department, University of Maryland, Baltimore County, Baltimore, MD 21250, USA
[3]NASA Goddard Space Flight Center, Code 616, Greenbelt, Maryland 20771, USA
[4]MS 475 NASA Langley Research Center, Hampton, VA 23681-2199, USA
[5]NASA Goddard Institute for Space Studies, New York, NY 10025, USA
[6]Netherlands Institute for Space Research (SRON, NWO-I), Utrecht, The Netherlands

**Correspondence:** Meng Gao (meng.gao@nasa.gov)

**Abstract.** NASA's Plankton, Aerosol, Cloud, ocean Ecosystem (PACE) mission, scheduled for launch in the timeframe of late 2022 to early 2023, will carry the Ocean Color Instrument (OCI), a hyperspectral scanning radiometer, and two multi-angle polarimeters (MAPs), the UMBC Hyper-Angular Rainbow Polarimeter (HARP2) and the SRON Spectro-Polarimeter for Planetary EXploration one (SPEXone). One purpose of the PACE MAPs is to better characterize aerosols properties, which can then be used to improve atmospheric correction for the retrieval of ocean color in coastal waters. Though this is theoretically promising, the use of MAP data in the atmospheric correction of collocated hyperspectral ocean color measurements has not yet been well demonstrated. In this work, we performed aerosol retrievals using the MAP measurements from the Research Scanning Polarimeter (RSP), and demonstrate its application to the atmospheric correction of hyperspectral radiometric measurements from SPEX Airborne. Both measurements were collected on the same aircraft from the Aerosol Characterization from Polarimeter and Lidar (ACEPOL) field campaign in 2017. Two cases over ocean with small aerosol loading (aerosol optical depth $\sim 0.04$) are identified including collocated RSP and SPEX Airborne measurements and Aerosol Robotic Network (AERONET) ground-based observations. The aerosol retrievals are performed and compared with two options: one uses reflectance only and the other use both reflectance and polarization. It is demonstrated that polarization information helps reduce the uncertainties of aerosol microphysical and optical properties. The retrieved aerosol properties are then used to compute the contribution of atmosphere and ocean surface for atmospheric correction over the discrete bands from RSP measurements and the hyperspectral SPEX Airborne measurements. The water leaving signals determined this way are compared with both AERONET and Moderate Resolution Imaging Spectroradiometer (MODIS) Ocean Color products with good agreement. The results and lessons-learned from this work will provide a basis to fully exploit the information from the unique combination of sensors on PACE for aerosol characterization and ocean ecosystem research.



## 1 Introduction

Ocean color remote sensing is a powerful tool for quantifying and monitoring global ocean ecosystems (Dierssen and Randolph, 2013), and provides valuable information for the estimation of phytoplankton biomass (Craig et al., 2012), primary productivity (Carr et al., 2006), and dissolved (Siegel et al., 2014) and particulate carbon pools (Fichot and Benner, 2011). Estimation of the

ocean color signal from the total at-sensor space-borne or air-borne measurement is known as atmospheric correction, which removes the radiometric contributions of the atmosphere and ocean surface (Wang, 2010; Mobley et al., 2016). Quantifying the effect of atmospheric aerosols is a primary challenge in the atmospheric correction (Frouin et al., 2019), due to their diversity of size, composition, and morphology, and associated variability in absorption and scattering properties (Remer et al., 2019a). In addition, aerosol deposition into ocean waters contributes to the availability of nutrients that modulate phytoplankton growth

and ultimately influence the trophic state of ocean ecosystems (Mahowald et al., 2005; Westberry et al., 2019). Furthermore, the ocean itself and the biological activity it supports may also be a source of aerosol (O'Dowd et al., 2002; McCoy et al., 2015; Croft et al., 2019). Better characterization of aerosol micro-physical and optical properties is expected to improve the retrieved ocean color signal and, therefore, the derived geophysical products that describe ocean ecosystems (PACE, 2018; Werdell et al., 2019).

Multi-angle polarimeters (MAPs), radiometers that measure spectral polarization states at multiple view angles, have been demonstrated to improve the retrieval performance of aerosol microphysical properties (Mishchenko and Travis, 1997; Chowdhary et al., 2001; Hasekamp and Landgraf, 2007; Knobelspiesse et al., 2012), including for applications over ocean waters (Jamet et al., 2019). A limited number of satellite missions carrying polarimetric payloads have been launched (Dubovik et al., 2019), including the Polarization and Directionality of the Earth's Reflectances (POLDER) instrument that was hosted on

Polarization and Anisotropy of Reflectances for Atmospheric Sciences Coupled with Observations from a Lidar (PARASOL; 2004–2013) and on the short-lived ADEOS and ADEOS-II missions (Tanré et al., 2011). Several more satellite missions with MAP instruments are planned to be launched in the time frame of 2022-2023, such as the European Space Agency (ESA)'s Multi-viewing Multi-channel Multi-polarisation Imager (3MI) on Meteorological Operational Satellite - Second Generation (MetOp-SG) (Fougnie et al., 2018), and the National Aeronautics and Space Administration (NASA) Multi-Angle Imager for

Aerosols (MAIA) (Diner et al., 2018), and the NASA Plankton, Aerosol, Cloud, ocean Ecosystem (PACE) mission (Werdell et al., 2019).

The PACE observatory will include two MAPs, the UMBC Hyper-Angular Rainbow Polarimeter-2 (HARP2) (Martins et al., 2018) and the SRON Spectro-Polarimeter for Planetary EXploration one (SPEXone) (Hasekamp et al., 2019), as well as its primary instrument, a hyperspectral scanning radiometer referred to as the Ocean Color Instrument (OCI). The OCI instrument

will provide continuous spectral measurement from the ultraviolet (340 nm) to near infrared (890 nm) with 5 nm resolution, plus a set of discrete shortwave infrared (SWIR) bands centered on 940, 1038, 1250, 1378, 1615, 2130, and 2260 nm. OCI will tilt $\pm 20°$ fore/aft, switching at the sub-solar point, to minimize viewing Sun glint. HARP2 is a wide field-of-view imager that measures polarized radiances at 440, 550, 670, and 865 nm, where the 670 nm band will be at 60 viewing angles and the other bands at 10 viewing angles. SPEXone is a narrow swath imager that performs multi-angle measurements at 5 viewing





angles of $0°$, $\pm 20°$ and $\pm 58°$ on ground, in a continuous spectral range spanning 385-770 nm with a resolution of 2-3 nm for intensity, and 10-40nm for polarization (Rietjens et al., 2019).

Through the combination of OCI and the two MAPs, the PACE mission provides a novel opportunity to bridge polarimetric and hyperspectral observations and advance the retrieval of both aerosol and ocean properties (Remer et al., 2019a, b; Chowd-
hary et al., 2019). Near infrared (NIR) or SWIR bands are often used to derive aerosol properties over ocean waters, and that approach has been implemented for the Hyperspectral Imager for the Coastal Ocean (HICO) (Ibrahim et al., 2018). The multiband atmospheric correction (MBAC) approach which utilizes channels in the NIR to SWIR has been proposed for PACE OCI (Ibrahim et al., 2019). With the PACE instruments, a more accurate retrieval of the aerosol properties can potentially be achieved using the MAP measurements, and the improved aerosol knowledge can then be applied to advance the accuracy of
atmospheric correction for OCI observations. This advancement would be especially valuable over coastal waters, where both aerosol and water optical properties are often complex. To date, there are only a few studies on performing atmospheric correction for hyperspectral radiometer using aerosol properties retrieved from the co-located MAP measurements. This is primarily due to the limited availability of co-located MAP and hyperspectral radiometer measurements over ocean. One such dataset is available from the North Atlantic Aerosols and Marine Ecosystems Study (NAAMES) field campaign in 2015, where both the
GEO-CAPE Airborne Simulator (GCAS) (a hyperspectral radiometer) and RSP were deployed. These datasets have been used to study the hyperspectral ocean color retrievals (Chowdhary et al., 2018).

In the fall of 2017, the Aerosol Characterization from Polarimeter and Lidar (ACEPOL) field campaign, a collaboration between NASA and Netherlands Institute for Space Research (SRON), was conducted with six passive and active instruments on the NASA ER2 high-altitude aircraft (Knobelspiesse and et al., to be submitted). These included four MAPs: airHARP (the
airborne version of HARP2 and HARP Cubesat (McBride et al., 2019)), AirMSPI (the Airborne Multiangle SpectroPolarimetric Imager) (Diner et al., 2013), SPEX Airborne (the airborne version of SPEXone )(Smit et al., 2019) and the RSP (Research Scanning Polarimeter) (Cairns et al., 1999), and two lidars: HSRL-2(the High Spectral Resolution Lidar-2) (Burton et al., 2015) and CPL (the Cloud Physics Lidar) (McGill et al., 2002). SPEX Airborne collects hyperspectral radiometry, and thus can be used as a proxy for OCI in developing hyperspectral ocean color algorithms. The co-located MAPs and hyperspectral SPEX
Airborne measurements are similar to the PACE payload, and thus provide a proxy dataset to evaluate the aerosol retrieval results from MAPs and the use of these retrieved aerosol properties for hyperspectral atmospheric correction. In this study, we build further on our previous work (Gao et al., 2018, 2019) and use the MAP measurements from RSP to conduct aerosol retrievals with its well documented measurement uncertainty analysis (Knobelspiesse et al., 2019), and apply the results to the atmospheric correction of the SPEX Airborne measurements. We identified two cases over the ocean from ACEPOL where
SPEX Airborne measurements and Aerosol Robotic Network (AERONET) ground-based observations are collocated with RSP.

In order to retrieve aerosol information from polarimetric measurements over the ocean, a number of advanced aerosol retrieval algorithms have been developed for both airborne and spaceborne MAPs, such as POLDER/PARASOL (Hasekamp et al., 2011; Dubovik et al., 2011, 2014), AirMSPI (Xu et al., 2016, 2019), SPEX Airborne (Fu and Hasekamp, 2018; Fu et al.,
2019; Fan et al., 2019), RSP (Chowdhary et al., 2005; Wu et al., 2015; Stamnes et al., 2018; Gao et al., 2018, 2019), and



Directional Polarimetric Camera (DPC)/GaoFen-5 (Wang et al., 2014; Li et al., 2018). In this study, we use the Multi-Angle Polarimetric Ocean coLor (MAPOL) retrieval algorithm, which is a joint aerosol and water-leaving radiance retrieval algorithm designed with the bio-optical models applicable to both open and coastal waters (Gao et al., 2018, 2019). MAP measurements from RSP include the VIS (visible), NIR, and SWIR bands, which are used for joint aerosol and water leaving signal retrievals.

The impacts of including polarization information in the retrieval of the aerosol properties are studied by comparing the results with inputs of reflectance only and that of both reflectance and polarization in the MAPOL algorithm. We will also discuss the retrieval algorithm stability in terms of the sensitivity of the retrieval parameters to their initial guesses, and compare with the uncertainty estimation based on error propagation (Knobelspiesse et al., 2012). The atmospheric correction using the aerosol properties from MAP retrievals is complementary to the approaches using the reflectance at NIR and/or SWIR bands to derive

aerosol properties for the atmospheric correction on hyperspectral radiometers (Ibrahim et al., 2018, 2019), and is especially advantageous in scenarios where the aerosol properties in the VIS or ultraviolet (UV) bands cannot be accurately extrapolated from measurements in the NIR-SWIR spectral range (Chowdhary et al., 2019).

The paper is organized into six sections: Sect. 2 describes the data used in the retrieval and validation of aerosol micro-physical properties and water leaving signals, Sect. 3 reviews the MAPOL retrieval algorithm and recent updates for application

to hyperspectral atmospheric correction, Sect. 4 and 5 present the retrieval results and discussion, and Sect. 6 summarizes the conclusions.

## 2   Data

During the ACEPOL field campaign, there are four flight tracks with clear skies over the AERONET USC_SEAPRISM site, located at [33.564N, 118.118W] and mounted on an oil platform roughly 18km away from the coast (Knobelspiesse and et al.,

to be submitted). This site is part of AERONET-OC, which uses special instruments that observe the water leaving radiance in addition to the atmospheric state (Zibordi et al., 2009). Of those four, we examined two cases in detail, as summarized in Table 1, with both RSP and SPEX Airborne measurements collocated with the AERONET measurements at the USC_SEAPRISM site. The two measurements are at the time of 2017/10/23 21:33, and 2017/10/25 21:07. Hereafter we will refer the two cases as Case 10/23 and Case 10/25. The locations and viewing geometries for both RSP and SPEX are specified in Fig. 2. Case 10/23

is close to the principal plane with a relative azimuth angle of $8.7°$; while Case 10/25 is almost perpendicular to the principal plane with a relative azimuth angle of $94.6°$. The two cases have similar solar zenith angles of $53.3°$ and $50.9°$.

RSP is the airborne version of the Aerosol Polarimetry Sensor for the NASA Glory mission that has been flown in multiple field campaigns since 1999 (Cairns et al., 1999). It is a multi-angle scanner measuring 152 viewing angles within $60°$ fore and aft of nadir in the along track direction, in 9 channels from VIS to SWIR (center wavelengths 410, 470, 555, 670, 865, 960,

1590, 1880, 2250 nm). SPEX Airborne is a hyperspectral imager with the spectral range of 400-800 nm. Its spectral resolution is 10-20 nm for degree of polarization (DoLP) and 2-3 nm for intensity. SPEX Airborne has 9 viewing angles (different from the 5 viewing angles of SPEXone) within the angular range of $112°(\pm56°)$. The SPEX measurements with wavelengths larger



than 750 nm are excluded from our analysis due to a grating order overlap issue in the data (Smit et al., 2019; Fu et al., 2019). The RSP and SPEX Airborne data files used in this study are listed in the Data Availability section.

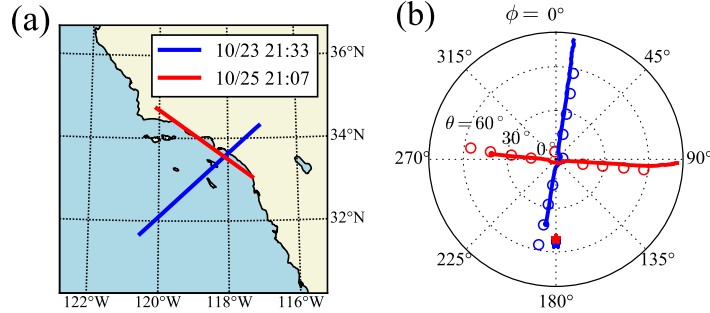

**Figure 1.** (a) The flight tracks across the AERONET USC_SEAPRISM site. The legend shows the time at which the aircraft flew over the AERONET site. (b) The corresponding polar plot for the RSP (solid line) and SPEX (open circles) viewing directions as summarized in Table 1, where $\theta$ is the zenith angle, and $\phi$ is the relative azimuth angle between the instrument viewing azimuth angle and the sunlight azimuth angle. The asterisk symbol indicate the antisolar point.

**Table 1.** Summary of the datasets from ACEPOL field campaigns used in this study.

| Date | 2017/10/23 | 2017/10/25 |
|---|---|---|
| UTC Time | 21:33 | 21:07 |
| Distance to AERONET site | 1.6 km | 1.2 km |
| Aircraft altitude | 20.1 km | 19.7 km |
| Solar zenith angle | 53.3° | 50.9° |
| RSP relative azimuth angle | 8.7° | 94.6° |
| RSP scattering angle range | [113.9°, 166.9°] | [108.5°, 129.7°] |

To validate the aerosol retrieval results, we will compare our retrieved aerosol properties with the aerosol products from HSRL-2 and AERONET. The HSRL instruments provide accurate assessment of AOD at 355 and 532 nm (Hair et al., 2008). In this study, the AOD product at 532 nm from the ACEPOL campaign with an assumed Lidar ratio is used. The AERONET USC_SEAPRISM site is equipped with a CIMEL based system called the Sea-Viewing Wide Field-of-View Sensor (SeaWiFS) Photometer Revision for Incident Surface Measurements (SeaPRISM), at eight wavelengths of 412, 443, 490, 532, 551, 667, 870, and 1020 nm with a bandwidth of 10 nm (Zibordi et al., 2009). The AERONET instrument provides direct Sun and diffuse sky-radiance measurements to infer aerosol properties (Holben et al., 1998). The measurements of direct Sun radiation are used to derive the spectral AOD with an uncertainty of 0.01 for mid-visible wavelengths and 0.02 for UV wavelengths (Holben et al., 1998; Eck et al., 1999; Holben et al., 2001; Smirnov et al., 2000). The diffuse sky-radiance measured at a wide range of scattering directions is used to infer aerosol size, complex refractive index and non-spherical particle ratio (Dubovik and King,





2000; Dubovik et al., 2006). For AOD less than 0.2, uncertainties in the AERONET inversion products are estimated as 15-35% for size distribution depending on aerosol types, 0.05 for refractive index and 0.05-0.07 for single scattering albedo (SSA) (Dubovik et al., 2000). The AERONET measurement capability can be extended to include photopolarimetric measurement with the next-generation Sun photometer, and its improvement on aerosol property retrievals has been demonstrated (Xu and

Wang, 2015; Xu et al., 2015; Fedarenka et al., 2016), although such retrievals were not available for this study.

The retrieved ocean color results are compared to AERONET Ocean Color (AERONET-OC) and the Moderate Resolution Imaging Spectroradiometer Ocean Color (MODIS OC) products. The MODIS OC product is processed with the atmospheric correction algorithm originating from Gordon and Wang (Gordon and Wang, 1994; Mobley et al., 2016) with the aerosol model from Ahmad et al. (Ahmad et al., 2010) and is publicly available (NASA Ocean Color Web). MODIS OC provides a

spatial coverage with 1km resolution at nadir. Since the aerosol properties and water leaving signals are measured or derived at different times, geometries and spatial resolutions, natural variation is a factor contributing to the difference when comparing the retrieved results. In order to evaluate the spatial variations when comparing with the retrieved water leaving reflectance, we averaged the MODIS (on board Aqua) water leaving reflectance within a 2km region around the USC_SEAPRISM site and compute its standard deviation as its maximum uncertainty. If smaller than 5%, the uncertainty is adopted as 5% which

is the accuracy goal for blue band and clear water (Hu et al., 2013). The AERONET measurements are available in almost every hour and there are a total of 8 measurements each day. The AERONET product provides good temporal coverage of the aerosol and ocean reflectance. We averaged the one-day AERONET products and compared its mean with the retrieval results, where the standard deviation (6% to 10% for both cases) is used to represent the maximum uncertainties. Note that the reported uncertainty for AERONET OC $R_{rs}$ is approximately 5% between 410-550nm(Zibordi et al., 2009).

The AODs for the two cases in our discussion are around 0.03∼0.04 at 550 nm, as reported by HSRL-2 and AERONET observations. It is challenging to retrieve aerosol micro-physical properties when the aerosol loading is small. Meanwhile the water leaving signals are often represented by the remote sensing reflectance ($R_{rs}$; $sr^{-1}$) as a ratio of the upwelling radiance and downwelling irradiance both just above the ocean surface (Mobley et al., 2016). In this study $R_{rs}$ is also small with a value around 0.002-0.003 $sr^{-1}$ from 400- 550 nm reported by the AERONET Ocean Color product, which account for 5%

to 15% of the total signal measured at the aircraft level. The percentage contribution of the water leaving reflectance to the observations depends on the polarization state and the water conditions (Chowdhary et al., 2012; Zhai et al., 2017). Although the aerosol loading is small, its contribution is of the same order of magnitude as the contribution of the water leaving signal contribution between 400-550 nm range, and hence remains important for atmospheric correction. Therefore, both the retrieval of aerosol micro-physical properties and the water leaving signals require high accuracy of the measurements from RSP and

SPEX Airborne.

Smit et al. provided a thorough comparison of the reflectance and polarization measurements between SPEX Airborne and RSP from ACEPOL over ocean, cloud and land scenes (Smit et al., 2019). For the ocean scenes, eight flight tracks were selected, and both the random and systematic difference between the two sensors were analyzed. Over the four RSP bands of 410, 470, 550, 670 nm, the random noise contribution to differences of reflectances are 2%, 2%, 2% and 4%. The systematic

differences are larger than the random differences at 410 and 470 nm, which are around 4% and 3%. For DoLP, the random





differences are 0.007, 0.005, 0.003, and 0.008 for the four bands from 410 to 670 nm; and systematic differences are either similar or smaller. This suggests that DoLP differences are dominated by random errors, while reflectances show systematic difference larger than the random noise at 410 and 470 nm. The systematic difference in reflectance also poses challenges in atmospheric correction for SPEX Airborne in the wavelengths between 410 and 470 nm. Polarization information shows a

higher level of agreement between the two sensors and therefore should be more strongly weighted in the retrieval algorithm.

## 3    Method

The MAPOL algorithm is designed to jointly retrieve the aerosol and water leaving signals from MAP data, which has been validated with the synthetic simulated data (Gao et al., 2018) as well as the RSP measurements from field campaigns (Gao et al., 2019). The retrieval algorithm minimizes the difference between the MAP measurements and the forward model sim-

ulations computed from a vector radiative transfer forward model (Zhai et al., 2009, 2010). Two ocean bio-optical models are implemented in MAPOL: one with Chlorophyll-a concentration as the single retrieval parameter applicable to open ocean optical properties, and the other with seven parameters applicable to complex coastal waters (Gao et al., 2019). In this study we perform retrievals near the USC_SEAPRISM site where waters are mostly clear so that the bio-optical model parameterized by Chlorophyll-a concentration is used.

15       In the MAPOL algorithm used in this study, the aerosol size distribution is composed by five sub-modes, each with a lognormal distribution with fixed mean radius and variance (Dubovik et al., 2006; Xu et al., 2016, 2017). The first three sub-modes (median radii of 0.1, 0.1732, 0.3 $\mu$m) represent the fine mode aerosols, and the last two sub-modes (median radii of 1.0, 2.9 $\mu$m) represents the coarse mode aerosols. For a general study, Fu and Hasekamp discussed the representation of aerosol size distribution through various numbers of sub-modes and found that similar five mode approach can provide good retrievals

for most aerosol parameters (Fu and Hasekamp, 2018). The aerosol refractive index spectra for both fine and coarse modes are approximated as $m(\lambda) = m_0 + \alpha_1 p_1(\lambda)$, where $m_0$ and $\alpha_1$ are fitting parameters, and $p_1(\lambda)$ is the first order of the principal components, computed from the dataset derived from Shettle and Fenn (1979), including spectral refractive indices of water, dust-like, biomass, industrial, soot, sulfate, water soluble and sea salt aerosols (d'Almeida et al., 1991; Wu et al., 2015). There are two sets of $m_0$ and $\alpha_1$ for the real and imaginary refractive index spectra, respectively, for both the fine and coarse modes.

This means that there are a total of 8 parameters for the refractive indices. In summary, the retrieval parameters include 5 volume densities (one for each sub-mode), 8 parameters for the refractive indices of fine and coarse modes, one parameter for wind speed, and the Chlorophyll-a concentration, with a total of 15 parameters. After the aerosol properties are retrieved from the discrete bands of the RSP data, the same parameters $m_0$ and $\alpha_1$ are used in the aforementioned refractive index spectral representation to calculate the aerosol refractive indices at the SPEX Airborne wavelengths, which are then used in the radiative

transfer model to compute the contribution of aerosols at the corresponding wavelengths.

The Stokes parameters, $L_t$, $Q_t$, and $U_t$, from RSP measurements are used to define the total measured reflectance $\rho_t = (\pi r^2 L_t)/(\mu_0 F_0)$ and total DoLP $P_t = \sqrt{Q_t^2 + U_t^2}/L_t$ where $F_0$ is the extraterrestrial solar irradiance, $\mu_0$ is the cosine of solar zenith angle, $r$ is the Sun-Earth distance in astronomical units. Circular polarization (Stokes parameter V), not measured by





any of the polarimeters in ACEPOL, is often ignored for atmospheric studies (Kawata, 1978). The cost function is used to quantify the difference between the measurement and the forward model simulation and is defined as:

$$\chi^2(\mathbf{x}) = \frac{1}{N}\sum_i \left( \frac{[\rho_t(i) - \rho_t^f(\mathbf{x};i)]^2}{\sigma_t^2(i)} + \frac{[P_t(i) - P_t^f(\mathbf{x};i)]^2}{\sigma_P^2(i)} \right) \tag{1}$$

where $\rho_t^f$ and $P_t^f$ denotes the total reflectance and DoLP simulated from the forward model; the state vector x contains retrieval parameters for aerosols and ocean; subscript $i$ stands for the indices of the measurements at different viewing angles and wavelengths; and N is the total number of the measurements used in the retrieval. All RSP bands are used in our retrievals except for the two water vapor absorption bands at 960 nm and 1880 nm. The total uncertainties of the reflectance and DoLP used in the algorithm are denoted as $\sigma_t$ and $\sigma_P$, which includes three components: the instrument measurement uncertainties as summarized in Knobelspiesse et al. (2019) (e.g. absolute radiometric and polarimetric characterization uncertainty averaged to 0.03 and 0.002 for the RSP instrument used in ACEPOL) , the variance from averaging nearby RSP pixels (the average of 5 consecutive pixels are used in this study, which corresponds to a surface pixel size of approximately 1km), and the forward model uncertainties estimated as 0.015 and 0.002 for the radiometric and polarimetric uncertainties, respectively(Gao et al., 2019). All these uncertinaties are added in the quadrature of $\sigma_t$ and $\sigma_P$ in Eq. 1 to represent the total uncertainties. The weight of the measurements in the cost function depends on the inverse square of $\sigma_t$ and $\sigma_P$. As will be discussed shortly, there are higher weight on the DoLP than reflectance in the cost function. Because DoLP has less dependency on the noise correlation between angles due to its definition as a ratio of two observations (Knobelspiesse et al., 2012), the noise correlation has been ignored in this study.

The relative uncertainties for reflectance and DoLP, denoted as $\sigma_t/\rho_t$ and $\sigma_P/P_t$, are defined as the ratio of the total uncertainties over the measurement, and summarized in Table 2. The magnitude of uncertainties often depends on the viewing angles as shown in the panels (c) and (d) of Figure 2 and 3. Table 2 shows the minimum value among the viewing angles at each band which corresponds the largest weight in Eq. 1 for the corresponding band. The value of $\sigma_t/\rho_t$ is 3.4% from 410 nm to 865 nm, and around 4% to 5% for the SWIR bands. $\sigma_P/P_t$ is between 0.4% to 1.8% for the bands from 410 nm to 865 nm, and for the SWIR bands, $\sigma_P/P_t$ is between 6.1% to 15.1%. The percentage uncertainties of the polarizations in the two SWIR bands further increases when the DoLP value decreases. We have tested the effects of the DoLP at the two SWIR bands on the aerosol retrieval and found that including them does not improve the retrieval accuracies. Moreover, the PACE MAPs do not include polarimetric SWIR measurements so the SWIR DoLPs are not used in our retrievals. In summary we use seven bands of $\rho_t$ and five bands of $P_t$ in our retrievals, and the corresponding cost function is denoted as $7\rho_t + 5P_t$. In order to understand the impacts of the polarization information, we also conducted the retrievals with only reflectance in the cost function which is denoted as $7\rho_t$.



**Table 2.** The minimum relative uncertainties of reflectance ($\rho_t$) and DoLP ($P_t$) for the RSP bands. The SWIR DoLPs denoted by asterisks are not used in the retrievals.

| Cases | Wavelength(nm) | 410 | 470 | 550 | 670 | 865 | 1590 | 2250 |
|-------|----------------|-----|-----|-----|-----|-----|------|------|
| 10/23 | $\sigma_t/\rho_t(\%)$ | 3.4 | 3.4 | 3.4 | 3.4 | 3.4 | 3.9 | 4.7 |
|       | $\sigma_P/P_t(\%)$ | 0.5 | 0.5 | 0.7 | 1.0 | 1.8 | 8.5* | 15.1* |
| 10/25 | $\sigma_t/\rho_t(\%)$ | 3.4 | 3.4 | 3.4 | 3.4 | 3.4 | 3.7 | 4.8 |
|       | $\sigma_P/P_t(\%)$ | 0.4 | 0.4 | 0.5 | 0.6 | 1.1 | 6.1* | 14.5* |

The use of oceanic sun glint from satellite measurements has been proposed and demonstrated to help aerosol absorption retrievals (Kaufman et al., 2002; Ottaviani et al., 2013). However, we tested the retrievals with the sun glint data included for Case 10/23 (as shown in Fig. 2) and found that the sun glint reflection cannot be modelled well using the isotropic Cox-Munk model (Cox and Munk, 1954) which depends on the wind speed only. This may be due to the high spatial resolution of 200m from RSP measurement or the insufficient representation with a scalar wind speed. Therefore, the sun glint region for only Case 10/23 is removed within an angle of $40°$ around the specular reflection direction of the direct solar light as indicated in Figure 3. There is no data removed by the glint mask for Case 10/25 due to its cross-principal plane geometry. The remaining range of scattering angles used in the retrieval are shown in Table 1.

The retrieved aerosol properties will be compared and validated with the AERONET products in the next section, and then used to conduct atmospheric correction for both RSP and SPEX airborne measurement. The resultant water leaving signals as represented by the remote sensing reflectance can be computed using the water leaving reflectance reaching the sensor $\rho^{Sensor}$ as

$$R_{rs} = \rho_w^{Sensor}/(\pi r^2 t_d t_u) \tag{2}$$

where $t_d$ is the downward transmittance of the solar irradiance to the surface, and $t_u$ is the upward transmittance of the water leaving radiance to the sensor(Gao et al., 2019). $\rho_w^{Sensor}$ represents the water leaving signals originating from scattering in the ocean, and can be derived from the atmospheric correction process by subtracting the reflectance contribution of atmosphere and ocean surface from the measurement at the aircraft (Gao et al., 2019). In order to compare the water leaving signals measured at different times derived from RSP, MODIS and AERONET instruments, the directional dependence of the water leaving reflectance is removed to obtain the signal at the nadir direction (Mobley et al., 2016).

After obtaining the aerosol properties from RSP retrievals, the atmospheric contributions including all the scattering and absorption process related to the aerosols, molecules and ocean surface, and the $t_d$ and $t_u$ transmittance are computed for the hyperspectral SPEX spectral bands with the vector radiative transfer model by Zhai et al. (Zhai et al., 2009, 2010, 2018) and subtracted from the hyperspectral SPEX Airborne measurement. The gas absorption in the radiative transfer simulation, including contributions from ozone, oxygen, water vapor, nitrogen dioxide, methane, and carbon dioxide, are accounted by using the US standard atmospheric constituent profiles (Anderson et al., 1986) but with scaled amount of water vapor, ozone and oxygen. The total ozone column density used to scale the ozone profile is obtained from the Modern-Era Retrospective





analysis for Research and Applications, Version 2 (MERRA-2) from NASA's Global Modeling and Assimilation Office (Gelaro et al., 2017). The total amounts of water vapor and oxygen are computed from minimizing the difference between measurement and simulated SPEX Airborne measurement over all the bands. The hyperspectral variations of gas absorption are incorporated in the radiative transfer simulation using a method similar to the double-k method (Zhai et al., 2018).

## 4 Results

The MAPOL retrieval algorithm was applied to the RSP measurements for the Cases 10/23 and 10/25. To evaluate the retrieval stability and uncertainties, 150 sets of random initial guesses for all the 15 retrieval parameters were used for the cost functions of both $7\rho_t + 5P_t$ and $7\rho_t$. Each parameter was varied within a boundary as specified in Gao et al. (2018). The minimum cost function value $\chi^2_{min}$ for Case 10/23 and 10/25 are 2.8 and 3.8 with the cost function of $7\rho_t + 5P_t$ and reduced to 0.6 and 0.4

with the cost function of $7\rho_t$ (polarization information not considered, all else the same). The minimum cost function value corresponds to the best aerosol retrievals, and the remaining residuals relate to the measurements which cannot be completely represented by the forward models.

Using the best aerosol retrievals corresponding to $\chi^2_{min}$ for $7\rho_t + 5P_t$, the reflectance and DoLP are simulated and compared with the measurements as shown in Fig. 2 and Fig. 3. The positive viewing zenith angles refer to the viewing angles on the glint side, and the negative viewing zenith angles refer to the sun side. The solid line is for the measurement data and the dashed line

is the simulation results using the retrieved aerosol and ocean properties for the reflectance ($\rho_t$) and DoLP ($P_t$) as plotted in panels (a) and (b). The total uncertainties as discussed in Section 2 are plotted in panels (c) and (d) for the reflectance and DoLP respectively. The percentage difference between measurements and fittings is plotted in panels (e) and (f). For Case 10/23, the viewing directions are close to the principal plane. The sun glint is indicated by the shaded area in Figs. 2, which were excluded

in the retrieval as discussed in Section 3. Using the retrieved aerosol properties, the reflectance and DoLP were also computed in the glint region. Although both the reflectance and DoLP measurements in the glint region were not considered in this study, the comparison of the measurement and simulated results indicates a better agreement of the DoLP than the reflectance, which can be explained by the fact that the DoLP from reflection on the ocean surface does not depend on the assumed distribution of surface slopes. For the remaining angles, the absolute residual larger than 10% in Figs. 2(c) and 2(d) are mostly associated

with the small values of the measurement, such as the SWIR bands for $\rho_t$, and the DoLP at viewing angles less than $-20°$. On average, the residual for both reflectance and DoLP are smaller than 6%. For Case 10/25, the viewing angles are almost perpendicular to the principal plane, the DoLP is always larger than 0.3, and the residuals for DoLP are even smaller than previous case with a value less than 2% on average. For the SWIR bands at the viewing angles between $-10°$ and $20°$, Fig 3(e) shows residuals larger than 10% at 1590 nm and 2250 nm, which indicates the measurements cannot be modelled by the

forward model. However, the difference is not observed in the DoLP comparisons in Fig. 3(f). Due to the presence of invalid measurements in the 2250 nm band between $-10°$ and $20°$, some measurement uncertainties in this portion are not computed as shown in Fig 3(c) and the corresponding measurements are not counted in the cost function.



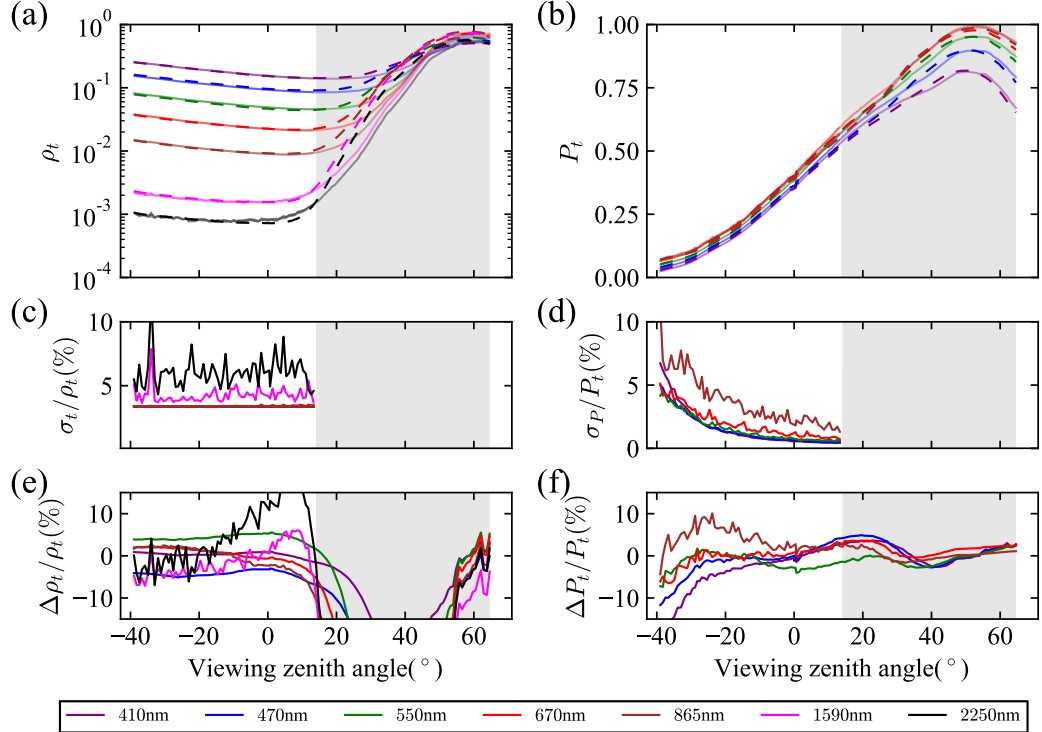

**Figure 2.** (**a**) and (**b**) The comparison of the RSP measurement and model fitting of reflectance $\rho_t$ and DoLP $P_t$ for Case 10/23, (**c**) and (**d**) are the total percentage uncertainties relative to the measurements for reflectance ($100\sigma_t/\rho_t$) and DoLP ($100\ \sigma_P/P_t$),(e) and (f) are the percentage residuals between the measurements and fittings relative to the measurements for reflectance ($100\Delta\rho_t/\rho_t$) and DoLP($100\Delta P_t/P_t$). The solid line in (a) and (b) is the measurement data and the dashed line is the simulation results from the retrieval. The shaded area indicates the angles not used in the retrieval (uncertainties are not calculated).





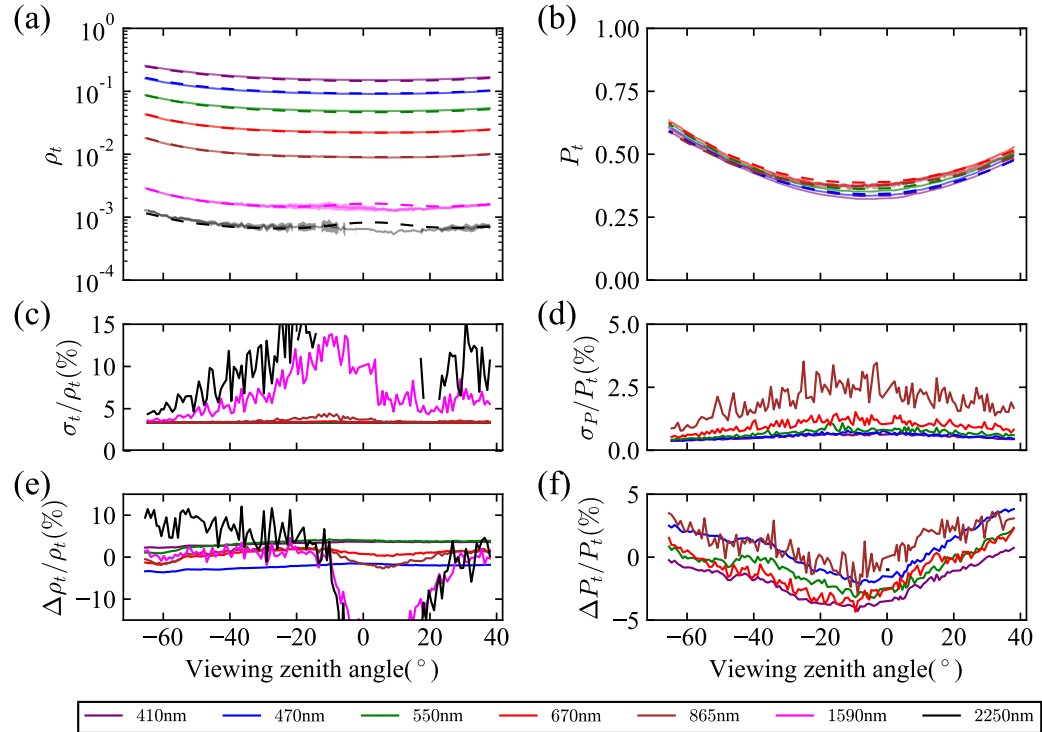

**Figure 3.** Same as Fig. 2 but for Case 10/25.

The histogram and cumulative probability of the cost functions for the 150 converged cases with random initial values are compared in Fig. 4. For the cost function $7\rho_t$ more than 50% cases are converged within a cost function value of 2 (the residual is sqrt(2) times the measurement uncertainty), while to have the same 50% cases converged for $7\rho_t + 5P_t$, the cost function value needs to be within 6. The wider spread of the cost function values is related to the higher sensitivity of DoLP in the

5    cost function, which is consistent with Table 2 showing that the DoLP has much smaller uncertainties. In order to evaluate the uncertainties of the best retrieval results, we consider the retrieval cases to be converged within a cut-off cost function value of $\chi_{min}^2 + 1$, and compute the standard deviation of all the aerosol properties and water leaving signals retrieved from these cases. This corresponds to the evaluation using all the retrieval cases converged within the cost function range of $[\chi_{min}^2, \chi_{min}^2 + 1]$. There are 50% of the retrievals in average converged within this cost function range for $7\rho_t$, while 30% of all the retrievals

10    converged within this requirement for $7\rho_t + 5P_t$. More discussion on the cut-off cost function value is in the next section. A comparison of the results with 75 and 150 cases demonstrated that the results are converged with enough samples to compute the standard deviation. The uncertainties due to the impact of the initial values is addition to the uncertainties from the error propagation method, where the measurement uncertainties are propagated to the retrieval parameters through the Jacobian matrix (e.g. Knobelspiesse et al. (2012)). A further discussion in the next section suggests that the uncertainties evaluated

15    using these two methods may also related.





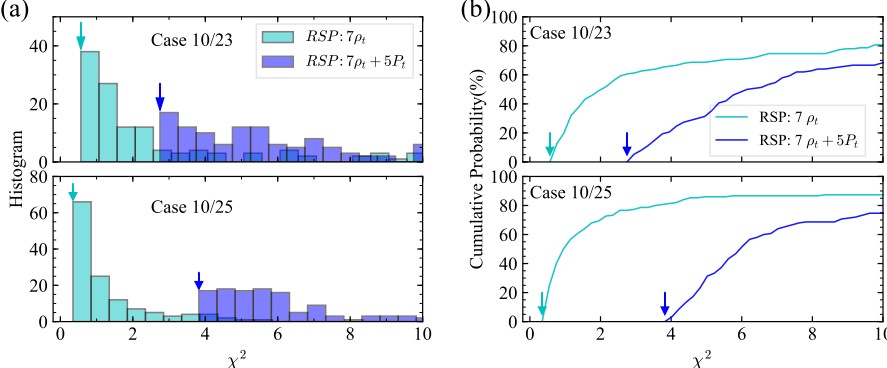

**Figure 4.** (a) The histogram for Case 10/23 and 10/25 with cost function $7\rho_t$ and $7\rho_t + 5P_t$ and a bin size of 0.5; (b) the corresponding cumulative probability. The arrows indicate the minimum cost function values $\chi^2_{min}$.

## 4.1 Aerosol micro-physical properties

We compared the retrieved aerosol size distribution, fine mode refractive index, fine mode SSA and total AOD with the averaged AERONET product on the same day in Fig. 5. The aerosol size distribution is represented by the volume density as a function of radius in the logarithmic scale. Both the shaded area and the error bar indicate 1-sigma uncertainty. The retrieval values and uncertainties are summarized in Table 3, which includes wind speed, refractive index, SSA and AOD for both fine and coarse modes, and remote sensing reflectance. The overall SSA for the two mode mixture is computed as the ratio of the number density weighted averages for the scattering and extinction cross sections (Bohren and Huffman, 1998). For Case 10/23, the retrieved wind speed for both cost functions are $3.4 \sim 3.5m/s$ with similar uncertainties. For 10/25, a wind speed of 5.4m/s is retrieved under $7\rho_t + 5P_t$ which is 1.5m/s faster than the results from $7\rho_t$ and with the uncertainty reduced by a factor of 2. Due to the exclusion of the glint in the retrieval for Case 10/23 and the cross principal plane geometry for Case 10/25, the wind is not ideal for retrieval. However, the retrieval uncertainties are within 0.5m/s for both cases with $7\rho_t + 5P_t$, which suggests that the measurements used in the retrievals are still influenced by wind speed.





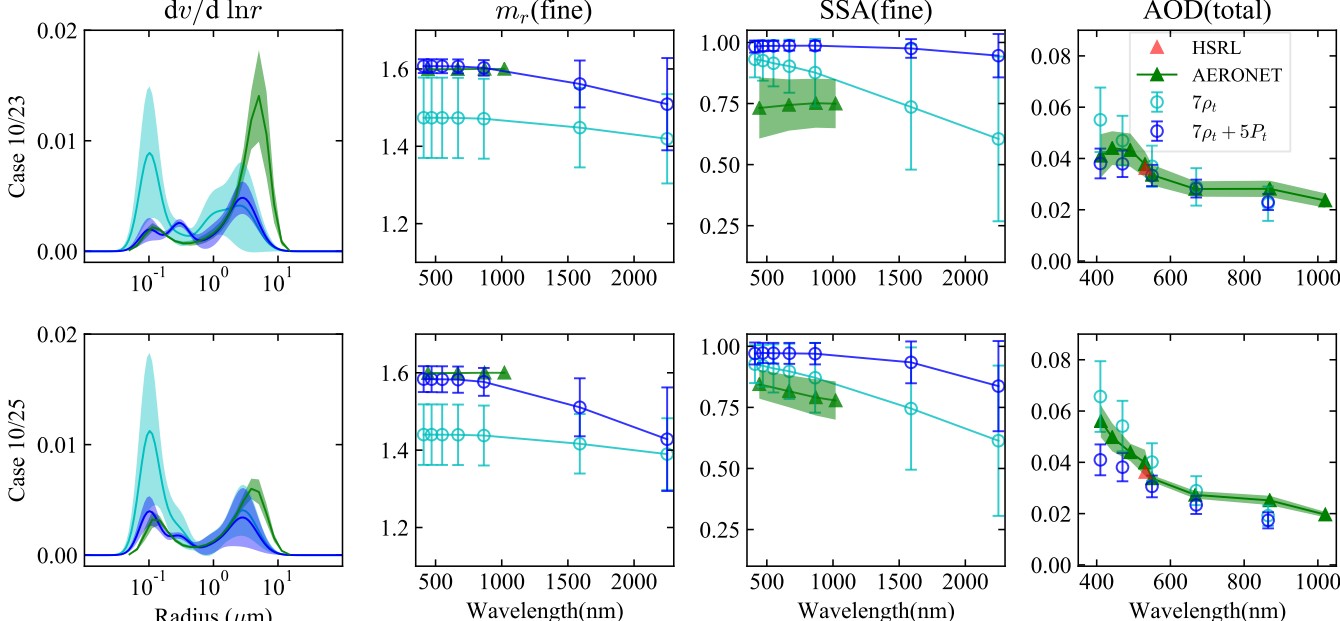

**Figure 5.** The aerosol size distribution ($dv/d\ln r$), fine mode refractive index($m_r$), fine mode SSA, and total AOD of the two cases on 10/23 and 10/25 retrieved with the cost functions of $7\rho_t + 5P_t$ and $7\rho_t$. The vertical width of the retrieved results indicates one sigma retrieval uncertainties. The results from AERONET product are plotted in green, and the vertical width indicates its daily variation. The HSRL AOD at 532 nm is indicated by the red triangle.





**Table 3.** The averaged retrieval values and uncertainties(in parenthesis) from the cases converged within the range of cost function $[\chi^2_{min},$ $\chi^2_{min} + 1]$ for fine and coarse mode effective radius ($r_{eff}$), effective variance ($v_{eff}$), refractive index ($m_r$), SSA, as well as total AOD and SSA, and remote sensing reflectance ($R_{rs}$). AOD, SSA and $m_r$ are all at 550 nm, and $R_{rs}$ is at both 470 and 550 nm.

| Date | 10/23 | 10/23 | 10/25 | 10/25 |
|---|---|---|---|---|
| Cost Function | $7\rho_t$ | $7\rho_t + 5P_t$ | $7\rho_t$ | $7\rho_t + 5P_t$ |
| Wind Speed(m/s) | 3.48(0.26) | 3.42(0.34) | 3.91(1.18) | 5.42(0.49) |
| $r_{eff}$(fine)($\mu m$) | 0.12(0.03) | 0.16(0.03) | 0.12(0.03) | 0.12(0.01) |
| $v_{eff}$(fine) | 0.26(0.11) | 0.45(0.08) | 0.28(0.09) | 0.41(0.05) |
| $m_r$(fine) | 1.47(0.10) | 1.61(0.02) | 1.44(0.08) | 1.58(0.03) |
| SSA(fine) | 0.91(0.09) | 0.99(0.02) | 0.91(0.10) | 0.97(0.04) |
| AOD(fine) | 0.025(0.005) | 0.027(0.004) | 0.035(0.007) | 0.026(0.003) |
| $r_{eff}$(coarse)($\mu m$) | 1.60(0.45) | 1.78(0.42) | 2.07(0.47) | 1.70(0.52) |
| $v_{eff}$(coarse) | 0.58(0.13) | 0.54(0.13) | 0.44(0.14) | 0.52(0.15) |
| $m_r$(coarse) | 1.52(0.12) | 1.56(0.11) | 1.57(0.10) | 1.53(0.11) |
| SSA(coarse) | 0.68(0.03) | 0.65(0.04) | 0.75(0.16) | 0.79(0.14) |
| AOD(coarse) | 0.012(0.007) | 0.006(0.003) | 0.005(0.003) | 0.005(0.003) |
| SSA(overall) | 0.84(0.07) | 0.92(0.03) | 0.89(0.09) | 0.94(0.04) |
| AOD(total) | 0.037(0.008) | 0.033(0.004) | 0.040(0.007) | 0.031(0.004) |
| $R_{rs}$(470nm)($sr^{-1}$) | 0.0027(0.0003) | 0.0026(0.0003) | 0.0024(0.0004) | 0.0025(0.0003) |
| $R_{rs}$(550nm)($sr^{-1}$) | 0.0021(0.0001) | 0.0020(0.0002) | 0.0020(0.0002) | 0.0021(0.0001) |

Fig. 5 shows that the aerosol size distribution retrieved with the polarized information $7\rho_t + 5P_t$ is closer to the AERONET results than that of the reflectance only retrieval ($7\rho_t$): the the maximum fine mode volume density reduced by 4 and 2.5 times for Case 10/23 and 10/25, and the uncertainties reduced by five times for both cases. The fine mode effective radii of $7\rho_t$ and $7\rho_t + 5P_t$ are similar for each, but the effective variances become larger when DoLP observations are included with an increase of around 0.1 to 0.2, as shown in Table 3. An increase of the fine mode effective variances can also be observed in Fig. 5 with a wider fine mode distribution. To compensate the much smaller fine mode density from the retrievals with $7\rho_t + 5P_t$, the retrieved fine mode refractive indices increase from 1.47 to 1.61 and from 1.44 to 1.58, for Cases 10/23 and 10/25 respectively. To compare with the AERONET results which assume that both fine and coarse modes have the same refractive index, we define the volume-averaged refractive index as $m_v = f_v \times m_r(\text{fine}) + (1 - f_v) \times m_r(\text{Coarse})$ where $f_v$ is the fine mode volume fraction (Hasekamp et al., 2011; Gao et al., 2018). For Cases 10/23 and 10/25 with $7\rho_t$, $m_v$ is 1.49 and 1.48 respectively. While with $7\rho_t + 5P_t$ $m_v$ becomes 1.58 and 1.56 for these two days which agree better with the AERONET refractive index of 1.6. Meanwhile, a larger fine mode SSA is also retrieved with the polarization information (from 0.91 to 0.99 for Case 10/23 and from 0.91 to 0.97 for Case 10/25) which suggests the aerosols have almost no absorption. The coarse mode SSAs are of $0.7 \sim 0.8$ for both days and both cost function options. Moreover, including polarization in the retrievals, the uncertainties for



refractive index, SSA and AOD become $0.02 \sim 0.03$ for refractive index, $0.02 \sim 0.04$ for SSA, and $0.004$ for AOD, which are reduced nearly by one half.

The AERONET SSAs at 550 nm are around 0.8 with a daily variation of 0.1, which are smaller than the retrieved overall SSA with a value of 0.92 for Case 10/23 and 0.94 for Case 10/25 with $7\rho_t + 5P_t$. The coarse mode SSAs are similar for both cost functions around 0.7 and 0.8 respectively, while the fine mode in RSP retrievals shows much less absorption. The HSRL AOD at 532nm, and the mean AERONET AOD at 550 nm are the same for Case 10/23 and 10/25 with a value of 0.036 and 0.034 respectively, which agrees well with the retrieved AODs (0.033 for Case 10/23 and 0.031 for Case 10/25). The AERONET AOD spectra are in good agreement with the RSP AOD spectra in terms of both shape and magnitude for Case 10/23, but are slightly larger than that of Case 10/25 with a difference of 0.011 at 410 nm. Note that AERONET aerosol product uncertainties are approximately 0.01 for AOD, 0.05 for refractive index, and 0.05-0.07 for SSA as mentioned in Section 2, which are comparable with the results for $7\rho_t$ but larger than the ones from $7\rho_t + 5P_t$. The differences between the AERONET aerosol product and aerosol retrievals with $7\rho_t + 5P_t$ are mostly within the uncertainty range. However, the aerosol signals are small due to the small AOD and can be easily influenced by other environmental factors not modeled by the forward model, more case studies are required to evaluate the overall agreement and bias.

## 4.2 Water leaving reflectance

The aerosol properties retrieved with the best fit results using $7\rho_t + 5P_t$ were applied to the atmospheric correction of the hyperspectral measurements from SPEX Airborne for both Case 10/23 and 10/25 using the methods discussed in Section 3. The retrieval uncertainties of the water leaving reflectance for the RSP $R_{rs}$ were computed similarly to that for the aerosol properties. The results are compared with the in-situ measurements from AERONET OC and MODIS OC products in Fig. 6. The AERONET and MODIS OC agree well with each other for Case 10/25; for Case 10/23 MODIS $R_{rs}$ is lower than AERONET $R_{rs}$ by a value of 0.001 $sr^{-1}$ at 470 nm. The uncertainties from MODIS OC and AERONET OC data as defined in Section 2 are smaller than the RSP retrieval results, which indicate the small temporal and spatial variations of the water leaving reflectance, and therefore their results can be used to compare with the RSP retrievals at a slightly different time and location (Table 1).





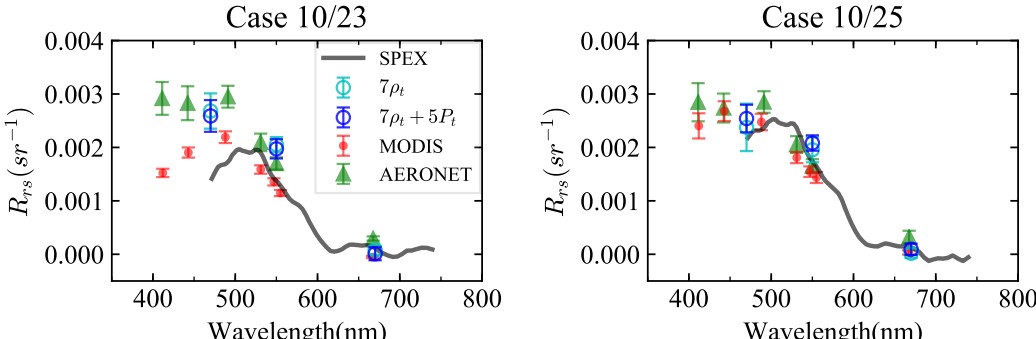

**Figure 6.** The remote sensing reflectance, $R_{rs}$, from MODIS OC, AERONET OC and the atmospheric corrections for RSP and SPEX Airborne for Case 10/23 and 10/25.

The RSP 410nm band is excluded from comparison due to the observation of 4% decrease in its radiometric throughput, while other RSP bands maintain stable within $\sim 1\%$ in the radiometric calibration and $\sim 0.1\%$ in the polarimetric calibration (Knobelspiesse and et al., to be submitted). The SPEX $R_{rs}$ at wavelengths shorter than 470 nm are not compared because of the observed absolute systematic difference with the RSP 410 and 470 nm bands as discussed in Section 2. The retrieved

RSP $R_{rs}$ using the aerosol properties from $7\rho_t + 5P_t$ and $7\rho_t$ shows similar values and uncertainties for both Case 10/23 and 10/25. The following discussions refer to RSP $R_{rs}$ results with cost function $7\rho_t + 5P_t$. Case 10/25 shows good agreement between SPEX Airborne, RSP, MODIS OC and AERONET OC. At 470 nm, the RSP and SPEX $R_{rs}$ are 0.0025 and 0.0022 $sr^{-1}$, and at 550 nm, they are 0.0021 and 0.0017 $sr^{-1}$ with a difference less than 0.0004 $sr^{-1}$. For Case 10/23, the SPEX $R_{rs}$ spectrum is consistent with the AERONET and RSP results for the wavelengths longer than 500 nm, while $R_{rs}$ from SPEX

is smaller than both AERONET and RSP at shorter wavelengths. At 470 nm, the RSP and SPEX $R_{rs}$ are 0.0026 and 0.0014 $sr^{-1}$ respectively, and the MODIS $R_{rs}$ is in between. The difference of RSP, SPEX and MODIS $R_{rs}$ at wavelengths smaller than 500 nm may be related to the measurement uncertainties where the effects are larger for the same percentage uncertainties due to the larger total measurement values. Another possible reason for the discrepancy between the MODIS $R_{rs}$ and others is the different aerosol models used for atmospheric correction. For MODIS, it is determined from the two NIR bands of 748

and 869 nm while others are based on polarimeter retrievals.

As shown in Fig. 6, for both Case 10/23 and 10/25, there are two small dips between 600 and 700nm in the remote sensing reflectance which are related to the oxygen absorption bands peaked at 688 nm (B-band) and 629 nm ($\gamma$ band). It is challenging to correct for the impacts from the strong gas absorption due to interaction between multiple scattering and gas absorption. In future studies any improvement requires knowledge of aerosol height and the exact instrument line shape function.

**5   Discussion**

Accurate determination of water leaving signals is key to derive ocean water optical properties and ocean biogeochemical conditions. In this work we have shown an example of how information rich observations of the atmosphere (from a MAP)



can be used to perform an atmospheric correction on highly spectrally resolved measurements of the ocean. As discussed in Section 2, there are systematic differences between RSP and SPEX Airborne at 410 nm and 470 nm with values of 4% and 3% respectively. The impact on the aerosol retrievals are likely mitigated by relying more on the polarization information with much smaller uncertainties than reflectance and better agreement between RSP and SPEX Airborne measurements, but the

computation of the water leaving signals cannot avoid the bias in reflectance. The atmospheric correction process requires the subtraction of the total measured reflectance by the simulated contributions from atmosphere and ocean surface. Therefore, the uncertainties and bias in the measurements can directly impact the water leaving signals retrievals. The reflectances $\rho_t$ measured are by RSP at 410 and 470 nm are 0.15 and 0.09, respectively. Based on the definition of $R_{rs}$, the 4% and 3% systematic difference in the reflectance will transfer into $R_{rs}$ biases around 0.002 and 0.0009 $sr^{-1}$. The random difference

between RSP and SPEX measurements at 470nm band is 2% as discussed by Smit et al. (2019) which can transfer to 0.0006 $sr^{-1}$ in $R_{rs}$. The differences of the $R_{rs}$ from RSP and SPEX at 470nm (0.0012 for Case 10/23 and 0.0003 for Case 10/25 with $7\rho_t + 5P_t$) may be due to the combined effects of the random and systematic differences in their measurements.

Accurate retrieval of $R_{rs}$ requires a higher level of measurement accuracy, especially for the shorter wavelengths where the total reflectance is more dominated by the contribution of Rayleigh scattering. To remove the bias in measurement and

improve atmospheric corrections, vicarious correction techniques using ground-based measurements has been developed for ocean color radiometry (Franz et al., 2007), and have also been applied to hyperspectral measurements (Ibrahim et al., 2018). Cross calibration between a MAP and hyperspectral radiometer will be critical for combining their results together for the purpose of ocean color remote sensing. Challenges in the measurement accuracy for ocean color observations will be addressed in the PACE mission, where the hyperspectral OCI measurement will achieve high calibration accuracies through pre-launch cal-

ibration campaigns, on-orbit gain adjustment through solar diffuser and lunar measurements, and on-orbit vicarious calibration (Werdell et al., 2019). On-orbit MAP cross-calibration with OCI will be possible – for example, measurements at the $\pm 20°$ viewing angle of SPEXone are expected to be cross-calibrated with OCI, transfering the high radiometric accuracy from OCI to SPEXone (Werdell et al., 2019)

Aerosol micro-physical properties are important to compute their optical properties and conduct atmospheric correction.

As shown in Table 3, the retrieved $R_{rs}$ for both Cases 10/23 and 10/25 is not sensitive to the exact aerosol micro-physical properties. Both $7\rho_t$ and $7\rho_t + 5P_t$ produce similar $R_{rs}$ and small relative uncertainties of $0.0003 \sim 0.0004 sr^{-1}$ at 470 nm. This is due to the small aerosol loading, the relatively simple aerosol micro-physical properties with almost no absorption, and the flat spectral dependency of the refractive index in VIS. A complete evaluation of the impact of the aerosol properties retrieved from MAP on atmospheric correction requires studies with more complex aerosol properties such as absorbing aerosols , which

will be a subject of study in the future.

Meanwhile, we have shown polarization information can help to improve accuracy in the retrieval of aerosol optical depth, fine mode refractive index and SSA as shown in Fig. 5. These reduced uncertainties in the aerosol micro-physical properties can help to determine aerosol type and its composition, and therefore provide valuable information in the study of aerosol deposition to the ocean and its impact on the ocean ecosystem and, potentially, the role of the ocean in aerosol formation. Based

on the aerosol properties retrieved for Case 10/23 and 10/25 as shown in Fig. 5, we derived an aerosol type from the aerosol





refractive index, SSA and extinction angstrom exponent(EAE) defined as EAE$=-\ln(AOD_{\lambda_1}/AOD_{\lambda_2})/\ln(\lambda_1/\lambda_2)$(Russell et al., 2014), where seven aerosol types are defined: pure dust, polluted dust, urban-industrial/developed economy, urban-industrial/developing economy, biomass burning/white smoke, biomass burning/dark smoke, and pure marine. Since the fine mode and coarse mode refractive indices are independent parameters in our retrievals, we perform the aerosol typing separately

for the fine and coarse modes. For the Case 10/23 retrieval after adding polarization, the fine mode refractive index at 550 nm increases from 1.47 to 1.61 with much reduced uncertainties, and the fine mode SSA increased from 0.91 to 0.99. The EAE computed using 470 and 865 nm for the fine mode AOD changes from 2.1 to 1.1. Following the typing scheme from Russell et al. (2014), with the polarization information, the fine mode particle could be interpreted as polluted dust, while with reflectance only results, the aerosol type may fall into the biomass burning type with higher EAE values and refractive index.

For coarse mode aerosols, we observe large uncertainties for the volume distribution and refractive index, and therefore it is challenging to determine its aerosol type accurately. The large uncertainties in coarse model aerosol properties may be due to the small contribution of the coarse mode signal in the total reflectance and the large total uncertainties in the SWIR DoLP (not used in retrievals).

Furthermore, the retrieval uncertainties evaluated in this study are for the retrievals converged within a range of cost function

value $[\chi^2_{min}, \chi^2_{min} + 1]$. This approach provides a statistical evaluation of the uncertainties relating to the cost function sensitivity and impact of initial values. When considering $[\chi^2_{min}, \chi^2_{min} + 3]$, 50% of data considered for $7\rho_t + 5P_t$ and 80% for $7\rho_t$ fall in this range as shown in Fig 4 (b), which lead to the uncertainties of fine mode refractive index increasing to $0.04 \sim 0.05$ for $7\rho_t + 5P_t$, still smaller than the uncertainties of $7\rho_t$ with a value around $0.08 \sim 0.1$ (Fig 7). The choice of cut-off cost function value used for retrievals in practice would depend on the accuracy requirement and the algorithm to determine the

initial values. The study by Knobelspiesse et al. (2012) estimated retrieval uncertainties using the error propagation method for the Aerosol Polarimetry Sensor (APS) (similar to RSP characteristics used in this study, although they used SWIR polarization) with various optical depths and aerosol types. The solar zenith and azimuth angles used by Knobelspiesse et al. (2012) are both $45°$, which are similar to the solar zenith angles $\sim 50°$ used in this study and in between of the two solar azimuth angles ($8.7°$ and $94.6°$) as shown in Table 1. Their results showed the uncertainties for fine mode refractive index, SSA and total AOD to

be 0.015, 0.02 and 0.005 when $AOD = 0.039$. The corresponding uncertainties evaluated in this study with a similar AOD are $0.02 \sim 0.03$, $0.02 \sim 0.04$ and 0.004 for fine mode refractive index, SSA and total AOD as shown in Table 3.

The uncertainty results computed from two different approaches are comparable to each other in magnitudes. The error propagation method directly relates the retrieval uncertainties to the measurement uncertainties by projecting them from measurement to state space using Jacobians calculated from the radiative transfer model. This method accurately represents retrieval

uncertainty if: (1) the measurement uncertainty is correct, (2) the forward model is an accurate and complete representation of physical reality, (3) the state space is locally linear about the retrieval, and (4) the retrieval algorithm is able to successfully converge to the smallest cost function value without artifacts. In practice, (2) is nearly always approximate, and (3) and (4) are not always the case, so the methodology used in Knobelspiesse et al. (2012) can be considered the best case retrieval uncertainty. It is, however, a convenient metric for retrieval uncertainty estimation since Jacobians are often calculated as part of the retrieval

process and can be reused for this purpose. The retrieval uncertainty method used in this work (expressing the volume in state



space containing cost function values calculated with many retrievals performed using randomly generated initial values) is similar in some respects, as it also relies on an accurate measurement uncertainty model (1) and forward model (2). However, it may be more accurate in some cases, since it does not make the assumption of local linearity (3) and can incorporate some potential convergence artifacts (4). Because this technique requires execution of many retrievals, it is computationally ineffi-

cient for operational use. It is, however, very relevant for a (data) limited study such as this, as it provides the best possible uncertainty estimate.

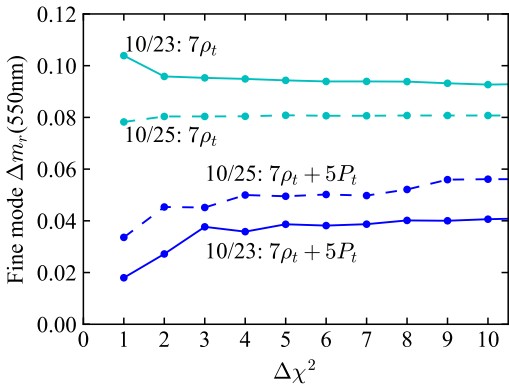

**Figure 7.** The uncertainties of the fine mode refractive index ($\Delta m_r$) at 550 nm computed from the cases converged within the cost function range of $[\chi_{min}^2, \chi_{min}^2 + \Delta\chi^2]$ where $\Delta\chi^2$ is from 1 to 10.

# 6   Conclusions

We have retrieved aerosol properties using multi-angle polarimetric measurement from RSP in the ACEPOL field campaign over the AERONET USC_SEAPRISM site. The aerosol properties are then applied to the hyperspectral SPEX Airborne mea-

surement to compute water leaving signals. This is a proof of concept for the application of MAP data on PACE to the atmospheric correction of the OCI spectrometer on the same mission. We demonstrated the improved accuracy when combining the reflectance and polarization in the retrievals compared to reflectance only retrievals. After adding DoLP in the retrieval cost functions, the uncertainties for aerosol refractive index, SSA and AOD are reduced by a factor of 2 for the two cases considered in the study. The absolute values also agree better with the AERONET aerosol product. Moreover, the higher ac-

curacy in DoLP measurements introduces larger weights of DoLP in the cost function relative to the reflectance measurement, and therefore provides resilience to the uncertainties and bias in the reflectance measurements, and produces a higher aerosol retrieval quality.

In order to apply the retrieved aerosol properties from the MAP measurements to hyperspectral atmospheric correction, the principal components of the aerosol refractive index spectra are interpolated into the bands specified for SPEX airborne. The

retrieval parameters from MAP measurements can be used directly with the hyperspectral measurements without interpolation. The resulting hyperspectral water leaving reflectances agree well with the AERONET OC and MODIS OC products.

The systematic differences between the RSP and SPEX Airborne radiometric measurements at 410 and 470 nm lead to larger discrepancies for the water leaving radiance from the SPEX airborne data. In the context of the PACE mission, the aerosol properties can be retrieved from the two PACE MAPs: SPEXone and HARP2, and applied to the hyperspectral measurement from the OCI instrument which has a much higher radiometric accuracy (although SPEXone and HARP2 have different char-

acteristics than RSP). With the MAPs and OCI on PACE, both aerosol micro-physical properties and hyperspectral ocean color signals will be obtained simultaneously with a global coverage and the knowledge will help the study, monitoring and protection of the ocean ecosystem.

*Data availability.* The data files for RSP, SPEX Airborne and and HSRL-2 used in this study are listed below. The RSP data is available from the NASA GISS website (NASA RSP Data Site), and the SPEX L1C and HSRL-2 files are available from the ACEPOL website(https:

//www-air.larc.nasa.gov/missions/acepol/index.html).

  – Case 10/23

   RSP: RSP2-ER2_L1B-RSPGEOL1B-GeolocatedRadiances_20171023T210750Z_V001-20171024T034314Z.h5

   SPEX: L1C-20171023_211047_150-213119_220_1000m_radiance.h5

   HSRL-2: ACEPOL-HSRL2_ER2_20171023_R1.h5

– Case 10/25

   RSP: RSP2-ER2_L1B-RSPGEOL1B-GeolocatedRadiances_20171025T204909Z_V001-20171026T030529Z.h5

   SPEX: L1C-20171025_204857_50-210929_170_1000m_radiance.h5

   HSRL-2: ACEPOL-HSRL2_ER2_20171025_R1.h5

*Author contributions.* MG generated the scientific data used in this paper and wrote the original manuscript. PWZ and BF formulated the

original concept for this study. MG and PWZ developed the retrieval algorithm. PWZ developed the hyperspectral radiative transfer code used in this study. KK and BC advised on the retrieval uncertainty evaluation and aerosol typing. AI, BF, and YH advised on the atmospheric correction. SC advised on the aerosol interaction with the ocean. PJW advised on the PACE instruments. OH and GF provided and advised on the SPEX Airborne data. All authors participated the writing and editing of this paper.

*Competing interests.* The authors declare no conflict of interest.

*Acknowledgements.* The authors would like to thank the ACEPOL team for conducting the field campaign and providing the data, thank the Oregon State University team for maintaining the AERONET USC_SEAPRISM site. Funding for the ACEPOL campaign from the Radiation Sciences Program managed by Dr. Hal Maring is acknowledged. Part of this work is funded by the NWO/NSO project ACEPOL: Aerosol Characterization from Polarimeter and Lidar under project number ALW-GO/16-09. M. Gao, B. Franz, B. Cairns, K. Knobelspiesse,



A. Ibrahim, S. Craig, and J. Werdell acknowledge support from the NASA PACE Project. P. Zhai acknowledges support from NASA Grant 80NSSC18K0345 under the Remote Sensing of Water Quality Program. M. Gao would like to thank Ivona Cetinić, Chris Proctor and Snorre Stamnes, Minwei Zhang for constructive discussions.





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
