# Peer review of "Inversion of multi-angular polarimetric measurements from the ACEPOL campaign: an application of improving aerosol property and hyperspectral ocean color retrievals"

_Atmospheric Measurement Techniques, 2020_

## Referee Comment (RC1) · Anonymous Referee #3 · 6 Apr 2020

The study aims at demonstrating the benefit of using synergistically hyperspectral and multi-angular polarimetric (MAP) observations to improve ocean color remote sensing, especially in the coastal zone, where aerosols are complex, relatively abundant, and highly variable. The approach is to use aerosol properties (size distribution parameters, index of refraction, optical thickness) retrieved from MAP data in a forward radiative transfer model to estimate the aerosol signal, therefore perform atmospheric correction of the hyperspectral measurements. To achieve this objective RSP and SPEX aircraft measurements acquired off the West Coast of California were used, and the

retrievals of aerosol properties and, therefore, remote sensing reflectance were compared with AERONET-OC measurements. Uncertainties in aerosol retrievals are reduced substantially (factor of 2) when using polarization and reflectance instead of just reflectance data, and the retrieved quantities show some agreement with in-situ measurements. The authors conclude that the findings constitute a proof-of-concept for the PACE mission, i.e., MAP data would be used in a similar way to correct atmospheric influence on the OCI hyperspectral imagery.

The approach is technically sound, the inversion techniques appropriate and robust, and the data processing/analysis performed carefully, but several issues prevent publication of the manuscript. First, aerosol abundance during the flights analyzed is very small, i.e., about 0.02 at 865 nm. With such minimum loadings, the signal to correct is so small that even large errors in the aerosol model would still yield sufficient accuracy on the remote sensing reflectance. It is not surprising, therefore, that even though differences are relatively large between estimates of size distribution, real part of index of refraction, and single scattering albedo using 7rhos and 7rhos + 5Pols (e.g., Figure 5), the retrieved RSP remote sensing reflectance is similar. I suspect that simply using the aerosol information from the MERRA-2 data would have provided similar performance. In other words, the demonstration is not credible when using cases with almost no aerosols. Second, HARP2 on the PACE mission will not measure in the shortwave infrared, so the demonstration should have been made using 5rhos and 5rhos + 5Ps to better mimic/represent the PACE capabilities. Furthermore, no comparison was made with remote sensing reflectance retrievals performed by the standard algorithm applied to aircraft RSP and SPEX data (possible even though for SPEX the spectral range is limited in the near infrared), in order to evaluate potential improvements by the proposed method. Finally, examining Figure 6, one cannot convincingly conclude that SPEX-derived hyperspectral reflectance in the blue agree with the in-situ measurements, i.e., in Section 4 the statement "The resulting hyperspectral water leaving reflectances agree well with the ARONET OC and MODIS OC products" in incorrect.

[Figure]

The above criticisms notwithstanding, the study is interesting. The procedures for estimating the atmospheric interference are well defined. I would recommend showing retrievals over the entire 2 flights (along and perpendicular to the coast) to capture varied aerosol and water reflectance situations, even though in situ measurements may not be available, compare the remote sensing reflectance retrievals with those of the standard algorithm, and evaluate against the aircraft lidar measurements and satellite products, but this would require a new submission.

———————————————

---

## Referee Comment (RC2) · Anonymous Referee #1 · 7 Apr 2020

The paper uses data collected from two airborne multi-angle polarimeters (MAPs) flying together on the ER-2 over a SeaPRISM site off the southern California coast to investigate whether multi-angle polarimetry will improve atmospheric correction of a hyperspectral instrument. The question is important because of the NASA PACE mission scheduled to launch in less than three years. The flagship PACE instrument is a hyperspectral radiometer, but it will be flying with two MAPs. Will those MAPs improve the radiometer's ability to retrieve ocean-leaving radiance by constraining aerosol properties? The study is presented well, is backed up with real validation and comes to a

solid conclusion. There are a few points that I think should be considered before publication, but overall my take is that the revisions will be very minor.

Comments:

1. Addressing the lack of UV in the study.

For me the biggest challenge for atmospheric correction in PACE is not the hyperspectral, but the UV. The atmosphere in the UV range is thick with Rayleigh and with aerosol scattering/absorption, making atmospheric correction even more uncertain than it is even in the deep blue (410 nm). Yet, the ocean community is excited by the UV measurements by OCI and intends to exploit that data, which they absolutely will not be able to do without a better plan for UV atmospheric correction.

I fully understand that addressing UV is outside the scope of this paper, but there are small things that can be done here to clarify the limitations of this paper and express the need for a future focus on the UV. The authors would be doing the community a great service.

P3 Line 1. SPEXone has true UV measurements.

P4 Line 27. SPEX airborne does not have UV measurements

P3 Lines 21-22. "SPEX Airborne collects hyperspectral radiometry, and thus can be used as a proxy for OCI in developing hyperspectral ocean color algorithms." With the caveat that it is missing measurements in the true UV part of the range.

P20 Line 2. "The resulting hyperspectral water leaving reflectances agree well with the AERONET OC and MODIS OC products." But not below 470 nm. This has implications for the UV.

2. Cases at very low aerosol loading

The two cases examined in the study are at very low AOD. There are a few places in the paper where the low aerosol loading introduces some concerns. P5 Line 11. "For

AOD less than 0.2, uncertainties in the AERONET inversion properties.... (Dubovik et al., 2000)".

For what wavelength is AOD < 0.2?

Dubovik 2000 is a very old reference. I looked through the materials on the AERONET web site including this document.

https://aeronet.gsfc.nasa.gov/new\_web/Documents/U27\_summary\_final.pdf

It seems to imply a different set of uncertainties that are actually larger than what is stated here, especially for refractive indices and SSA. Size distribution products can tolerate lower aerosol loading, but anything to do with absorption just falls apart when there is insufficient signal.

Also the implication by this statement on P5 is that the same uncertainties hold for all AOD 0.2 and less. This means that AOD = 0.04 has the same uncertainties as AOD = 0.20, and the AERONET document, and especially the graphs at the bottom do not support this.

Now I find it interesting that the authors do mention the challenge of retrieving microphysical properties when the aerosol loading is small. (P6 Line 15 and P16 Lines 6-8). Then why imply minimal uncertainty for these very, very low loadings of the cases studied in this paper? I am sufficiently distressed about trying to make retrievals of intrinsic particle properties when the AOD at 550 nm is less than 0.04, that I question the results of these retrievals in Figure 5 and Table 3, and some of the overall conclusions. Specific items that need to be addressed:

The statement on P5 should be updated.

In figure 5, the AERONET properties are plotted with their daily variation, but not their uncertainties. The uncertainties are larger than the range of the daily variation. This should be stated explicitly.

P16 Line 4, check those AERONET uncertainties again. I think they are too small.

P19 Line 16. The absolute values also agree better with the AERONET aerosol product. "This is not true for SSA. But... there are large uncertainties for SSA in AERONET because of the low AOD.

3. Systematic biases between airborne SPEX and RSP

Beginning on P6 Line 28, the paper mentions "systematic differences", but never describes which is higher, SPEX or RSP. This becomes important in the conclusion. P17 Lines 15-18. Here systematic differences between RSP and SPEXairborne are mentioned again, but which one is higher? And it isn't stated which one is right. So when speaking of impact on aerosol retrievals, how would these instrumental differences cascade into the aerosol retrievals? What should be expected from these differences, and why or why not were these expectations met? P17 Lines 21-23. Once again we are presented with differences, but are never told which instrument produces the higher result.

4. Theoretical retrieval uncertainty and validation against measurements

The authors to their credit address uncertainty from both the theoretical perspective and then also by comparing with ground-based measurements with well-defined uncertainty. The act of validation validates the magnitudes of the retrieved properties. In addition the act of validation validates the theoretical estimate of the uncertainties. The authors should explain explicitly in the paper when the theoretical polarimeter uncertainty is validated by the ground-based measurements and when it is not. I never believe the calculations of theoretical uncertainty until validation. On P17 Lines 11-12, "These reduced uncertainties in the aerosol micro-physical properties can help to determine aerosol type and its composition.." The authors here are discussing the theoretical reduced uncertainties. Have these reduced uncertainties been explicitly validated?

The concern I have is that the authors believe their theoretical calculations of uncertainty too much. P16 Lines 3-5. "Note that AERONET aerosol product uncertainties are approximately 0.01 for AOD, 0.05 for refractive index, and 0.05-0.07 for SSA as mentioned in Section 2, which are comparable with the results for 7rho\_t but larger than the ones from 7rho\_t +5rho\_t. " The implication is that RSP is more accurate than AERONET. There might be argument that RSP SSA retrievals are more accurate than AERONET inversion, but there is no way the RSP retrievals of AOD are going to be better than the AERONET direct sun measurements.

The point is that theoretical uncertainty calculations can only calculate the uncertainty that is known and when a retrieval is made in the real world, then the uncertainty that cannot be quantified theoretically, enters the picture and the actual accuracy of the retrieval is less good than the theoretical calculation.

5. The results question the ability of PACE to produce Rrs at short wavelengths

P20 Line 2. "The resulting hyperspectral water leaving reflectances agree well with the AERONET OC and MODIS OC products." Well, not towards the blue, near 470 nm. Hasn't this always been the problem? Ocean biology products need water leaving radiance at the short wavelengths, and they are going to want the UV also. Here the authors show that towards the blue, the hyperspectral retrieved water leaving radiance deviates from AERONET and MODIS products by a lot. At 470 nm for the 10/23 case, the SPEX airborne retrieved remote sensing reflectance is half of what AERONET-OC measured. This does not bode well for the ability to use the blue and UV from the PACE hyperspectral Ocean Color Instrument in any reliable, consistent fashion. The authors examine two cases. In one out of the two cases the atmospheric correction fails at shorter wavelengths for the hyperspectral retrieval. This needs to be stated explicitly when describing Figure 6, but also explicitly in the Conclusions.

6. Smaller items, but some are still substantial

Abstract Line 4. "aerosols properties" should be "aerosol properties"

P6 Line 19. "water leaving reflectance". Is the same as Remote Sensing reflectance mentioned on Line 16? The terms seemed to be used interchangeably, and I'm not sure that is correct.

Figure 6 caption. What do the error bars signify?

P8 Line 8. "uncertainties" is misspelled.

P9 Line 9. Does the rho\_Sensor need a 'w' subscript?

P9 Lines 13-15. That statement, "\_Sensor w represents the water leaving signals originating from scattering in the ocean, and can be derived from the atmospheric correction process by subtracting the reflectance contribution of atmosphere and ocean surface from the measurement at the aircraft (Gao et al., 2019)." This statement warrants an explicit equation so that the reader does not need to look up the reference. Maybe repeat from the Mobley reference also.

P9 Lines 26-27. "The total amounts of water vapor and oxygen are computed from minimizing the difference between measurement and simulated SPEX Airborne measurement over all the bands. " This statement could be expanded upon to provide greater clarity.

P9 Line 32. "Each parameter was varied within a boundary as specified in Gao et al. (2018)." Could we have the details repeated here? The authors draw heavily upon references to their previous publications, which is fine, but these details need to be repeated here to make this paper complete in its own right.

P10 Lines 5-6. "viewing angles on the glint side, and the negative viewing zenith angles refer to the sun side." Isn't the glint and the sun on the same side? Glint is forward scattering. This confusion continues throughout. The paper needs this clarified.

Table 3 caption. "parenthesis" should be parentheses. Plural.

P15 Lines 10-12. "The coarse mode SSAs are of 0:7 \_ 0:8 for both days and both cost

function options. Moreover, including polarization in the retrievals, the uncertainties for refractive index, SSA and AOD become 0:02 \_ 0:03 for refractive index, 0:02 \_ 0:04 for SSA, and 0:004 for AOD, which are reduced nearly by one half." Because these two sentence run one after the other, the second sentence appears to refer to the coarse mode, but the numbers seem to represent the conditions of the fine mode.

P16 Line 13. "in situ measurements". The MODIS retrievals are certainly not in situ measurements, and it is debatable whether we should be calling the AERONET SeaPRISM measurements "in situ". Possibly for SeaPRISM.

P17 Lines 5-6. "The difference of the MODIS and SPEX Rrs at wavelengths smaller than 500 nm may be related to the measurement uncertainties where the effects are larger for the same percentage uncertainties due to the larger total measurement values." I did not understand this sentence at all.

P18 Line 7. VIS is never previously defined. Just write out "visible"

P18 Line 10-11. "Meanwhile, we have shown polarization information can help to improve accuracy in the retrieval of aerosol optical depth, fine mode refractive index and SSA as shown in Fig. 5." I actually see the opposite in Figure 5 for SSA, at least the retrieval without polarization gets closer to AERONET retrievals, but really how can we believe any of it when AOD is less than 0.04?

P18 Lines 21-23. Do the authors really believe this? I find it very far-fetched that they are trying to assign type to an aerosol with AOD less than 0.04. Really? The Russell study was using a data base where the entries all had significant loading. Whatever they found would have no relationship to the cases of the present study, because the present study is way outside of the Russell study's dynamic range. This speculation should just be removed from the paper.

Figure 7 is never referenced in the text.

P19 Line 16. "The absolute values also agree better with the AERONET aerosol product." Not true for SSA.

P19-20 Lines 20-2. "In order to apply the retrieved aerosol properties from the MAP measurements to hyperspectral atmospheric correction, the principal components of the aerosol refractive index spectra are interpolated into the bands specified for SPEX airborne. The retrieval parameters from MAP measurements can be used directly with the hyperspectral measurements without interpolation." The two sentences are contradictory. The first states that the refractive index spectra have to be interpolated into hyperspectral. The second states that no interpolations is necessary.

---

## Author Comment (AC1) · 4 Jun 2020

Thanks for the comments and suggestions. Please find our response in the attached pdf file.

Please also note the supplement to this comment:
https://www.atmos-meas-tech-discuss.net/amt-2020-11/amt-2020-11-AC1-supplement.pdf

---

## Author Response (AR1)

The paper uses data collected from two airborne multi-angle polarimeters (MAPs) flying together on the ER-2 over a SeaPRISM site off the southern California coast to investigate whether multi-angle polarimetry will improve atmospheric correction of a hyperspectral instrument. The question is important because of the NASA PACE mission scheduled to launch in less than three years. The flagship PACE instrument is a hyperspectral radiometer, but it will be flying with two MAPs. Will those MAPs improve the radiometer's ability to retrieve ocean-leaving radiance by constraining aerosol properties? The study is presented well, is backed up with real validation and comes to a solid conclusion. There are a few points that I think should be considered before publication, but overall my take is that the revisions will be very minor.

Thanks for the interest in this work and the positive comments.

Comments:
1. Addressing the lack of UV in the study.
For me the biggest challenge for atmospheric correction in PACE is not the hyperspectral, but the UV. The atmosphere in the UV range is thick with Rayleigh and with aerosol scattering/absorption, making atmospheric correction even more uncertain than it is even in the deep blue (410 nm). Yet, the ocean community is excited by the UV measurements by OCI and intends to exploit that data, which they absolutely will not be able to do without a better plan for UV atmospheric correction.

I fully understand that addressing UV is outside the scope of this paper, but there are small things that can be done here to clarify the limitations of this paper and express the need for a future focus on the UV. The authors would be doing the community a great service.

Thank you for the comments. We agree that the UV coverage which will be provided by PACE is important for both atmosphere and ocean community, but unfortunately it cannot be explored in this study due to the limitation in the measurement data. We have made necessary revisions to emphasize the importance of atmospheric correct at UV, which are listed below in the responses to the specific comments.

P3 Line 1. SPEXone has true UV measurements.
P4 Line 27. SPEX airborne does not have UV measurements
P3 Lines 21-22. "SPEX Airborne collects hyperspectral radiometry, and thus can be used as a proxy for OCI in developing hyperspectral ocean color algorithms." With the

caveat that it is missing measurements in the true UV part of the range.

We added the following sentence to clarify the missing UV measurements in SPEX Airborne:

> "The spectral range from the SPEX Airborne measurements used in this study is from 470 to 750 nm. This does not cover the UV bands, which is nevertheless important and deems further research for the PACE mission (Frouin et al 2019, Chowdhary et al 2019)."

P20 Line 2. "The resulting hyperspectral water leaving reflectances agree well with the AERONET OC and MODIS OC products." But not below 470 nm. This has implications for the UV.

We revised the statement by specifying more details for both RSP Rrs and SPEX Rrs:

> "…
> The retrieval uncertainties on RSP Rrs is within 0.0004 $sr^{-1}$ ( same to SPEX Rrs), while the comparison of the two cases with the AERONET Rrs shows a difference less than 0.0003 $sr^{-1}$ for RSP Rrs, and a maximum difference of 0.0004 $sr^{-1}$ (Case 10/25) and 0.001 $sr^{-1}$ (Case 10/23) for SPEX Rrs. The difference of SPEX Rrs for Case 10/23 is larger than the retrieval uncertainties which is likely due to the radiometric uncertainties from the sensors.
> "

2. Cases at very low aerosol loading
The two cases examined in the study are at very low AOD. There are a few places in the paper where the low aerosol loading introduces some concerns. P5 Line 11. "For AOD less than 0.2, uncertainties in the AERONET inversion properties…. (Dubovik et al., 2000)".
For what wavelength is AOD < 0.2?

Dubovik 2000 is a very old reference. I looked through the materials on the AERONET web site including this document.
https://aeronet.gsfc.nasa.gov/new_web/Documents/U27_summary_final.pdf
It seems to imply a different set of uncertainties that are actually larger than what is stated here, especially for refractive indices and SSA. Size distribution products can tolerate lower aerosol loading, but anything to do with absorption just falls apart when there is insufficient signal.

Also the implication by this statement on P5 is that the same uncertainties hold for all AOD 0.2 and less. This means that AOD = 0.04 has the same uncertainties as AOD =0.20, and the AERONET document, and especially the graphs at the bottom do not support this.

Now I find it interesting that the authors do mention the challenge of retrieving microphysical properties when the aerosol loading is small. (P6 Line 15 and P16 Lines 6-8).

Then why imply minimal uncertainty for these very, very low loadings of the cases studied in this paper? I am sufficiently distressed about trying to make retrievals of intrinsic particle properties when the AOD at 550 nm is less than 0.04, that I question the results of these retrievals in Figure 5 and Table 3, and some of the overall conclusions.

Thank you for the comments on the AOD, also thank you for providing the AERONET uncertainty document. We agree that the statement on the AERONET aerosol uncertainties may not best describe the particular cases in our study. We revised our manuscript according to the reviewer's suggestions, which are listed below in the responses of specific comments.

Specific items that need to be addressed:

The statement on P5 should be updated.

It is challenging to provide an accurate theoretical assessment of the uncertainties for the particular cases in our studies, we removed the general statement, and instead we only use the uncertainties evaluated by the daily average of the AERONET product as an estimate. We also made it explicit that the inversion uncertainties increase with smaller AOD by citing the AERONET uncertainty document:

> "Note that the actual inversion uncertainties for the aerosol properties, such as the refractive index and single scattering albedo (SSA), may be larger than their daily averaged result for small AOD cases as reported by the AERONET Version 3 uncertainty analysis (Description of Aerosol Inversion Uncertainty for Level 2 Products). In general, AERONET retrievals of aerosol microphysical properties become less certain as AOD decreases."

In figure 5, the AERONET properties are plotted with their daily variation, but not their uncertainties. The uncertainties are larger than the range of the daily variation. This should be stated explicitly.

We specifically mentioned that the AERONET results are plotted with its daily variation as:
> "The results from AERONET product are plotted in green, and the vertical width indicates its daily variation."

The possible larger uncertainty from AERONET product are also added as in the response to the last comment.

P16 Line 4, check those AERONET uncertainties again. I think they are too small.
This is similar to the above statement in P5, we removed it.

P19 Line 16. The absolute values also agree better with the AERONET aerosol product. " This is not true for SSA. But… there are large uncertainties for SSA in AERONET because of the low AOD.

We revised it as follows:

> "The absolute values also agree better with the AERONET aerosol product except SSA probably due to the large uncertainty of SSA from both the MAP and AERONET inversions at low AOD."

3. Systematic biases between airborne SPEX and RSP
Beginning on P6 Line 28, the paper mentions "systematic differences", but never describes which is higher, SPEX or RSP. This becomes important in the conclusion. P17 Lines 15-18. Here systematic differences between RSP and SPEXairborne are mentioned again, but which one is higher? And it isn't stated which one is right. So when speaking of impact on aerosol retrievals, how would these instrumental differences cascade into the aerosol retrievals? What should be expected from these differences, and why or why not were these expectations met?

Thank you for the questions. The systematic differences the reviewer referred are the differences in radiometric calibration between the two instruments as reported by Smit et al 2019. The radiometric bias from both instruments could contribute to the systematic difference between these two instruments. We have revised the paragraph to indicate which sensor has larger reflectance as follows:

> "…Over the four RSP bands of 410, 470, 550, 670 nm, the random noise contribution to differences of reflectance are 2%, 2%, 2% and 4%. RSP reflectance is slightly larger than SPEX reflectance at 410 and 470 nm as indicated by their systematic differences of around 4% and 3% respectively, larger than the random differences; the systematic differences at the other two bands are relatively small with values of 0% and 1%."

From the comparison of the Rrs with AERONET Rrs, it seems the radiometric calibration of RSP is more accurate at 410 and 470 nm (as in the revised statement in the response of comment 1). But it is not our intention to claim which measurement is more accurate than the other based on only two case studies. Moreover, in terms of aerosol retrievals, we are trying to mitigate the issues by relying more on the RSP DoLP measurements which has smaller systematic difference between instrument and also high sensor accuracy.

P17 Lines 21-23. Once again we are presented with differences, but are never told which instrument produces the higher result.

Similar to the response to the previous comment, we have added specifically which instrument produces higher results:

> "As discussed in Section 2, reflectance measured by RSP is larger than SPEX Airborne measurement by a systematic difference of 4% and 3% at 410 and 470 nm respectively."

Meanwhile in the same paragraph, we showed how much impact on the Rrs could come from the systematic difference on the bands of 410 and 470nm, and why we excluded the wavelength less than 470nm in Rrs comparisons. We added more discussions here:

> "The reflectances measured by RSP at 410 and 470 nm are 0.15 and 0.09, respectively. Based on the definition of Rrs, the 4% and 3% systematic difference in the reflectance will transfer into a large Rrs biases around 0.002 and 0.0009 sr$^{-1}$. Therefore the Rrs from both RSP and SPEX at wavelengths less than 470nm are not compared…"

4. Theoretical retrieval uncertainty and validation against measurements
The authors to their credit address uncertainty from both the theoretical perspective and then also by comparing with ground-based measurements with well-defined uncertainty. The act of validation validates the magnitudes of the retrieved properties. In addition the act of validation validates the theoretical estimate of the uncertainties. The authors should explain explicitly in the paper when the theoretical polarimeter uncertainty is validated by the ground-based measurements and when it is not. I never believe the calculations of theoretical uncertainty until validation. On P17 Lines 11-12, "These reduced uncertainties in the aerosol micro-physical properties can help to determine aerosol type and its composition.." The authors here are discussing the theoretical reduced uncertainties. Have these reduced uncertainties been explicitly validated?

Thank you for the comments on validations. To make our discussion more accurate, we revised the corresponding statement by referring to explicitly "retrieval accuracy", and "retrieval uncertainties", and added the limitation in validation data and the requirement of future validation campaigns:

> "Meanwhile, we have shown polarization information can help to improve retrieval accuracy in the retrieval of aerosol optical depth, fine mode refractive index and SSA as shown in Fig. 5. Besides the theoretical retrieval accuracy analysis, validations with direct measurements are important to account for unknown uncertainties. The AOD results from polarimetric retrievals can be validated with ground-based measurement such as AERONET and lidar measurements such as HSRL, however, it is challenging to validate complex aerosol refractive index, SSA, and size distribution for the entire atmospheric column due to the lack of direct measurements. Such validation requires well-planned airborne field campaigns, concepts for which are under development (PACE validation plan 2020)"

The concern I have is that the authors believe their theoretical calculations of uncertainty too much. P16 Lines 3-5. "Note that AERONET aerosol product uncertainties are approximately 0.01 for AOD, 0.05 for refractive index, and 0.05-0.07 for SSA as mentioned in Section 2, which are comparable with the results for 7rho_t but larger than the ones from 7rho_t +5rho_t . " The implication is that RSP is more accurate than AERONET. There might be argument that RSP SSA retrievals are more accurate

than AERONET inversion, but there is no way the RSP retrievals of AOD are going to be better than the AERONET direct sun measurements.

The point is that theoretical uncertainty calculations can only calculate the uncertainty that is known and when a retrieval is made in the real world, then the uncertainty that cannot be quantified theoretically, enters the picture and the actual accuracy of the retrieval is less good than the theoretical calculation.

This is a very good point. We revised our manuscript as in the response to previous comments with acknowledgement of the limitation of the theoretical uncertainty and the necessary for more direct validation, which is repeated as follows:

> "Meanwhile, we have shown polarization information can help to improve retrieval accuracy in the retrieval of aerosol optical depth, fine mode refractive index and SSA as shown in Fig. 5. Besides the theoretical retrieval accuracy analysis, validations with direct measurements are important to account for unknown uncertainties. The AOD results from polarimetric retrievals can be validated with ground-based measurement such as AERONET and lidar measurements such as HSRL, however, it is challenging to validate complex aerosol refractive index, SSA, and size distribution for the entire atmospheric column due to the lack of direct measurements. Such validation requires well-planned airborne field campaigns, concepts for which are under development (PACE validation plan 2020)"

5. The results question the ability of PACE to produce Rrs at short wavelengths P20 Line 2. "The resulting hyperspectral water leaving reflectances agree well with the AERONET OC and MODIS OC products." Well, not towards the blue, near 470 nm. Hasn't this always been the problem? Ocean biology products need water leaving radiance at the short wavelengths, and they are going to want the UV also. Here the authors show that towards the blue, the hyperspectral retrieved water leaving radiance deviates from AERONET and MODIS products by a lot. At 470 nm for the 10/23 case, the SPEX airborne retrieved remote sensing reflectance is half of what AERONET-OC measured. This does not bode well for the ability to use the blue and UV from the PACE hyperspectral Ocean Color Instrument in any reliable, consistent fashion. The authors examine two cases. In one out of the two cases the atmospheric correction fails at shorter wavelengths for the hyperspectral retrieval. This needs to be stated explicitly when describing Figure 6, but also explicitly in the Conclusions.

Thank you for the comments. In the response to comment 1 (repeat below), we revised statement on the comparison of RSP, SPEX and AERONET Rrs in the conclusion:

> "…
> The retrieval uncertainties on RSP Rrs is within $0.0004$ sr$^{-1}$ (same to SPEX Rrs ), while the comparison of the two cases with the AERONET Rrs shows a difference less than $0.0003$ sr$^{-1}$ for RSP Rrs, and a maximum difference of $0.0004$ sr$^{-1}$ (Case 10/25) and $0.001$ sr$^{-1}$ (Case 10/23) for SPEX Rrs. The difference of SPEX Rrs for Case 10/23 is larger than

the retrieval uncertainties, which is likely due to the radiometric uncertainties from the sensors.
"

Meanwhile, MAP radiometric measurements are expected to have higher agreement with PACE OCI through cross-calibration, as discussed in section 5(second paragraph):

"…On-orbit MAP cross-calibration with OCI will be possible – for example, measurements at the $\pm 20^o$ viewing angle of SPEXone are expected to be cross-calibrated with OCI, transfering the high radiometric accuracy from OCI to SPEXone (Werdell et al 2019)"

The following discussion are added in the conclusion to clarify the implication to PACE OCI:

"Although the hyperspectral atmospheric correction for wavelength less than 470nm cannot be demonstrated by the SPEX airborne data, the PACE OCI will provide high quality hyperspectral measurement from 340 to 890nm and a few SWIR bands, and the demonstration of the atmospheric correction including UV spectral range will require future studies. "

More detailed comparison of the Rrs spectrum has been provided in the discussion of Figure 6

"… The RSP Rrs at 470 and 550nm are 0.0026 and 0.0020 respectively for Case 10/23, and 0.0025 and 0.0021 respectively for Case 10/25 as shown in Table 3. For AERONET Rrs, the values at 442, 490 and 550nm are 0.0027, 0.0028, 0.0017 sr$^{-1}$ for Case 10/23, and 0.0028, 0.0029, 0.0017 sr$^{-1}$ for Case 10/25. Using the interpolated value of AERONET Rrs at RSP bands, the difference between RSP and AERONET Rrs are within 0.0003 sr$^{-1}$."

More discussions on the comparison of SPEX and AERONET Rrs are in the revised file and diff file.

6. Smaller items, but some are still substantial
Abstract Line 4. "aerosols properties" should be "aerosol properties"

Done

P6 Line 19. "water leaving reflectance". Is the same as Remote Sensing reflectance mentioned on Line 16? The terms seemed to be used interchangeably, and I'm not sure that is correct.

Thanks for potting this out. We revised the "water leaving reflectance" as "water leaving signals". The definition of the water leaving reflectance is provided in Eq(2), and its connection with Rrs is in Eq(1).

Figure 6 caption. What do the error bars signify?

We revised the caption as follows:

> "The error bars for the RSP retrieved results with cost functions of 7ρt and 7ρt+ 5Pt indicate one sigma retrieval uncertainties. SPEX Airborne atmospheric correction use the same RSP retrieved aerosol models and therefore shares the same retrieval uncertainties (not indicated in plot). The error bar for the AERONET OC Rrs indicates its daily variation."

P8 Line 8. "uncertainties" is misspelled.
Corrected.

P9 Line 9. Does the rho_Sensor need a 'w' subscript?
Yes, thank you for spotting this. Corrected.

P9 Lines 13-15. That statement, "_Sensor w represents the water leaving signals originating from scattering in the ocean, and can be derived from the atmospheric correction process by subtracting the reflectance contribution of atmosphere and ocean surface from the measurement at the aircraft (Gao et al., 2019)." This statement warrants an explicit equation so that the reader does not need to look up the reference. Maybe repeat from the Mobley reference also.

Thanks for the suggestion. We added a formula for it:

"The water leaving reflectance $\rho_w^{Sensor}$ represents the signals originating from scattering in the ocean and reached the sensor, and can be derived from the atmospheric correction process as
$\rho_w^{Sensor} = \rho_t - \rho_{t,atms+sfc}^{Sensor}$
where $\rho_{t,atms+sfc}^{Sensor}$ is the reflectance contribution of atmosphere and ocean surface at the aircraft (Mobley et al 2016, Gao et al 2019)"

P9 Lines 26-27. "The total amounts of water vapor and oxygen are computed from minimizing the difference between measurement and simulated SPEX Airborne measurement over all the bands. " This statement could be expanded upon to provide greater clarity.

Thank you for the suggestion. We revised the statement as follows:

> "We then simulated the reflectance spectra under SPEX geometries with the retrieved aerosol properties and various amounts of oxygen and water vapor. The simulated spectra are compared with SPEX Airborne measurement, and the best amounts of water vapor and oxygen are chosen to minimize the difference between the measurements and simulations. During this process the aerosol properties and ozone density are kept unchanged. "

P9 Line 32. "Each parameter was varied within a boundary as specified in Gao et al. (2018)." Could we have the details repeated here? The authors draw heavily upon references to their previous publications, which is fine, but these details need to be repeated here to make this paper complete in its own right.

We revised the sentence to provide the range of the key parameters:

"Each parameter was varied within a boundary as specified in Gao et al. (2018 and 2019), where the wind speed is less than 10 m/s, the Chlorophyll a concentration is less than 30 mg/m$^3$, the aerosol refractive index varies effectively between 1.3 to 1.6 in its real part and between 0 to 0.03 in its imaginary part, and random mixing fractions of the five aerosol volume densities constrained by a maximum total AOD of 0.3."

P10 Lines 5-6. "viewing angles on the glint side, and the negative viewing zenith angles refer to the sun side." Isn't the glint and the sun on the same side? Glint is forward scattering. This confusion continues throughout. The paper needs this clarified.

We revised the Figure 1 captain by pointing out that the asterisk symbol indicate the antisolar point, we then revised the above statement as :

"…the negative viewing zenith angles refer to the antisolar point in Figure 1b.
… the reflectance and DoLP are simulated and compared with the measurements as shown in Fig. 2 and Fig. 3, where the viewing zenith angles are the same as defined in Figure 1(b) with the positive sign referring to the glint side ($\phi < 90^o$ or $\phi > 270^o$), and the negative sign referring to the other hemisphere containing the antisolar point."

Table 3 caption. "parenthesis" should be parentheses. Plural.
Corrected

P15 Lines 10-12. "The coarse mode SSAs are of 0.7 – 0.8 for both days and both cost function options. Moreover, including polarization in the retrievals, the uncertainties for refractive index, SSA and AOD become 0.02 – 0.03 for refractive index, 0.02 – 0.04 for SSA, and 0.004 for AOD, which are reduced nearly by one half." Because these two sentence run one after the other, the second sentence appears to refer to the coarse mode, but the numbers seem to represent the conditions of the fine mode.

Thank you for the comments. The sentences are revised as follows:

"For the fine mode, when including polarization in the retrievals, the uncertainties become 0.02 – 0.03 for refractive index, 0.02 – 0.04 for SSA, and 0.004 for AOD, with most values reduced by more than one half. The uncertainties for most coarse mode properties remain with similar magnitudes."

P16 Line 13. "in situ measurements". The MODIS retrievals are certainly not in situ measurements, and it is debatable whether we should be calling the AERONET

SeaPRISM measurements "in situ". Possibly for SeaPRISM.

Thanks for the suggestion. We revised the sentence as:

> "The results are compared with the MODIS OC products and the SeaPRISM measurements from AERONET OC in Fig.6"

P17 Lines 5-6. "The difference of the MODIS and SPEX Rrs at wavelengths smaller than 500 nm may be related to the measurement uncertainties where the effects are larger for the same percentage uncertainties due to the larger total measurement values." I did not understand this sentence at all.

We revised the sentences as follows:

> "The larger difference of RSP, SPEX and MODIS Rrs at wavelengths smaller than 500 nm may be related to the measurement uncertainties where the reflectance are larger at shorter wavelengths. "

P18 Line 7. VIS is never previously defined. Just write out "visible"
Done.

P18 Line 10-11. "Meanwhile, we have shown polarization information can help to improve accuracy in the retrieval of aerosol optical depth, fine mode refractive index and SSA as shown in Fig. 5." I actually see the opposite in Figure 5 for SSA, at least the retrieval without polarization gets closer to AERONET retrievals, but really how can we believe any of it when AOD is less than 0.04?

Here we intended to discuss the retrieval uncertainties, and we revised the sentences as follows:

> "Meanwhile, we have shown polarization information can help to reduce retrieval uncertainties in the retrieval of aerosol optical depth, fine mode refractive index and SSA as shown in Table. 3, but the retrieval accuracies are limited by the low AOD."

We also revised the statement in the conclusion to mention the disagreement with AERONET SSA:

> "The absolute values also agree better with the AERONET aerosol product except SSA probably due to the large uncertainty of SSA from both the MAP and AERONET inversions at low AOD".

P18 Lines 21-23. Do the authors really believe this? I find it very far-fetched that they are trying to assign type to an aerosol with AOD less than 0.04. Really? The Russell study was using a data base where the entries all had significant loading. Whatever they found would have no relationship to the cases of the present study, because the

present study is way outside of the Russell study's dynamic range. This speculation should just be removed from the paper.

Thanks, we removed the discussion on aerosol typing.

Figure 7 is never referenced in the text.
Figure 7 is referred in page 19.

P19 Line 16. "The absolute values also agree better with the AERONET aerosol product." Not true for SSA.

The sentence is revised as follows (as mentioned previously):

"The absolute values also agree better with the AERONET aerosol product except SSA probably due to the large uncertainty of SSA from both the MAP and AERONET inversions at low AOD".

P19-20 Lines 20-2. "In order to apply the retrieved aerosol properties from the MAP measurements to hyperspectral atmospheric correction, the principal components of the aerosol refractive index spectra are interpolated into the bands specified for SPEX airborne. The retrieval parameters from MAP measurements can be used directly with the hyperspectral measurements without interpolation." The two sentences are contradictory. The first states that the refractive index spectra have to be interpolated into hyperspectral. The second states that no interpolations is necessary.

Thank you for the comments. The interpolation is referred to the principal components of the refractive index spectra, each principal component is a spectrum. We removed the second sentence since the principal component coefficients cannot be interpolated.

Response to the report of reviewer #3:

We appreciate the detailed comments and suggestions from the reviewer, which are very helpful in improving the clarity of this work. Please find our responses with corresponding revisions below.
The study aims at demonstrating the benefit of using synergistically hyperspectral and multi-angular polarimetric (MAP) observations to improve ocean color remote sensing, especially in the coastal zone, where aerosols are complex, relatively abundant, and highly variable. The approach is to use aerosol properties (size distribution parameters, index of refraction, optical thickness) retrieved from MAP data in a forward radiative transfer model to estimate the aerosol signal, therefore perform atmospheric correction of the hyperspectral measurements. To achieve this objective RSP and SPEX aircraft measurements acquired off the West Coast of California were used, and the retrievals of aerosol properties and, therefore, remote sensing reflectance were compared with AERONET-OC measurements. Uncertainties in aerosol retrievals are reduced substantially (factor of 2) when using polarization and reflectance instead of just reflectance data, and the retrieved quantities show some agreement with in-situ measurements. The authors conclude that the findings constitute a proof-of-concept for the PACE mission, i.e., MAP data would be used in a similar way to correct atmospheric influence on the OCI hyperspectral imagery.

The approach is technically sound, the inversion techniques appropriate and robust, and the data processing/analysis performed carefully, but several issues prevent publication of the manuscript. First, aerosol abundance during the flights analyzed is very small, i.e., about 0.02 at 865 nm. With such minimum loadings, the signal to correct is so small that even large errors in the aerosol model would still yield sufficient accuracy on the remote sensing reflectance. It is not surprising, therefore, that even though differences are relatively large between estimates of size distribution, real part of index of refraction, and single scattering albedo using 7rhos and 7rhos + 5Pols (e.g., Figure 5), the retrieved RSP remote sensing reflectance is similar. I suspect that simply using the aerosol information from the MERRA-2 data would have provided similar performance. In other words, the demonstration is not credible when using cases with almost no aerosols.

Thanks for the summary and the positive comments in our approach and analysis. Also thanks for the discussions on the small AOD in our study. Here we provide more clarifications here:

1) We agreed that the aerosol loadings in these two cases are small which is about 0.03-0.04 at 550nm, and 0.02-0.03 at 865 as the reviewer corrected pointed out. However, due to the small value of the remote sensing reflectance, accurate retrieval of the aerosol properties is still important to determine the remote sensing reflectance. As verified from radiative

transfer simulations with aerosols only, aerosol reflectance contributes to the same order of magnitude as the remote sensing reflectance at 400-550nm range. The following discussions are provided and revised in the Section 2 (second last paragraph):

> "…Although the aerosol loading is small, its contribution is of the same order of magnitude as the water leaving signal between 400-550 nm range, and hence remains important for atmospheric correction. Therefore, both the retrieval of aerosol micro-physical properties and the water leaving signals require high accuracy of the measurements from RSP and SPEX Airborne."

2) We agree that the similar remote sensing reflectance obtained using the aerosol properties obtained from the two different cost function may relate to the small optical depth, and several other factors. Please note that the available cases with co-located multi-angle polarimetric measurement and hyperspectral measurements are really rare. We revised our manuscript to state the importance for future studies and validation campaigns with various aerosol loading. Some discussions and revisions are provided in Section 5 (third paragraph):

> "Meanwhile, we have shown polarization information can help to improve retrieval accuracy in the retrieval of aerosol optical depth, fine mode refractive index and SSA as shown in Fig. 5. Besides the theoretical retrieval accuracy analysis, validations with direct measurements are important to account for unknown uncertainties. The AOD results from polarimetric retrievals can be validated with ground-based measurement such as AERONET and lidar measurements such as HSRL, however, it is challenging to validate complex aerosol refractive index, SSA, and size distribution for the entire atmospheric column due to the lack of direct measurements. Such validation requires well-planned airborne field campaigns, concepts for which are under development (PACE validation plan 2020)"

3) Thank you for suggesting the use of MERRA2 aerosol model for atmospheric correction. We conducted the following evaluations:
   a. We located the corresponding MERRA2 one hour aerosol product as archived in https://oceandata.sci.gsfc.nasa.gov/ for the two cases in our study at 2017/10/23 21:33 and 2017/10/25 21:07 (file names are N201729621_AER_MERRA2_1h.nc and N201729821_AER_MERRA2_1h.nc).
   b. We then located the MERRA2 pixels near the AERONET SeaPRISM site location and found that the corresponding MERRA2 AOD are 0.054, and 0.080 at 550nm for cases 10/23 and 10/25. Note that AERONET AOD at 550nm are 0.034, while HSRL AOD at 532nm are 0.036 similar to both days. Our retrieved AODs at 550nm are 0.033 for Case 10/23 and 0.031 for Case 1025. The MERRA2 AOD overestimate AERONET AOD by 0.02 and 0.046 respectively.
   c. We estimated the amount of aerosol contribution to the remote sensing reflectance ($\Delta Rrs$) by using the single scattering approximation, namely, $\Delta Rrs \sim AOD \times B/\pi$, where B is the aerosol backscattering fraction at 550nm (around 0.2 for our cases). We have $\Delta Rrs \sim 0.001$ and 0.003 for Case 10/23 and 10/25 respectively, larger than the difference (<0.0005) between our retrievals and the AERONET Rrs at 550nm, especially for case 10/25.

    d.   Therefore, we concluded that MERRA2 aerosol model is not ideal for the atmospheric correction in our studied cases. However, the suggestion to investigate MERRA2 aerosol model is interesting. We can study whether MERRA2 aerosol model can be used as a way to better select initial values in the retrieval algorithm in our future study.

Second, HARP2 on the PACE mission will not measure in the shortwave infrared, so the demonstration should have been made using 5rhos and 5rhos + 5Ps to better mimic/represent the PACE capabilities.

Thank you for the suggestion on removing SWIR bands in aerosol retrievals. In this study we aim to provide best retrievals using the full capability of the RSP sensors, although DoLP measurements in SWIR bands are not used due to issues discussed in the manuscript. We did not intended to make RSP measurements to look the same as HARP or SPEX. Even after we removed the RSP SWIR bands, RSP are still different than HARP and SPEX with many more viewing angles and different measurement uncertainties. However, the algorithm and procedures using the current RSP measurements can be applied to other polarimetric measurements as a proof of concept demonstration to assist hyperspectral atmospheric correction. Furthermore, there are SWIR measurement in PACE OCI, which may have higher SNR, and it is potentially can be used to assist the MAP retrievals. We revised our manuscript as follows:

"The percentage uncertainties of the polarizations in the two SWIR bands further increases when the DoLP value decreases. We have tested the effects of the DoLP at the two SWIR bands on the aerosol retrieval and found that including them does not improve the retrieval accuracies, so the SWIR DoLPs are not used in our retrievals. Moreover, the PACE MAPs do not include polarimetric SWIR measurements but PACE OCI includes several SWIR bands measured at a single viewing angle and may have higher accuracy, a synergy of PACE OCI SWIR with MAP measurements may further improve aerosol retrievals."

Furthermore, no comparison was made with remote sensing reflectance retrievals performed by the standard algorithm applied to aircraft RSP and SPEX data (possible even though for SPEX the spectral range is limited in the near infrared), in order to evaluate potential improvements by the proposed method.

This is another good suggestion to apply standard atmospheric correction algorithm on RSP and SPEX data. However, this requires generating appropriate aerosol lookup table for the exact RSP and SPEX bands which are not currently available in the processing software. This suggestion deservers a separate study which is out of scope of this work.

Finally, examining Figure 6, one cannot convincingly conclude that SPEX-derived hyperspectral reflectance in the blue agree with the in-situ measurements, i.e., in Section 4 the statement "The resulting hyperspectral water leaving reflectances agree well with the ARONET OC and MODIS OC products" in incorrect.

Thank you for the comments. We made the following revisions to provide more details:

> "…
> The retrieval uncertainties on RSP Rrs is within 0.0004 sr$^{-1}$(same to SPEX Rrs), while the comparison of the two cases with the AERONET Rrs shows a difference less than 0.0003 sr$^{-1}$for RSP Rrs, and a maximum difference of 0.0004 sr$^{-1}$ (Case 10/25) and 0.001 sr$^{-1}$ (Case 10/23) for SPEX Rrs. The difference of SPEX Rrs for Case 10/23 is larger than the retrieval uncertainties which is likely due to the radiometric uncertainties from the sensors.
> "

We have also added more discussions on the difference between RSP Rrs, SPEX Rrs and AERONET. On the comparison between RSP and AERONET Rrs:

> "… The RSP Rrs at 470 and 550nm are 0.0026 and 0.0020 respectively for Case 10/23, and 0.0025 and 0.0021 respectively for Case 10/25 as shown in Table 3. For AERONET Rrs, the values at 442, 490 and 550nm are 0.0027, 0.0028, 0.0017 sr$^{-1}$ for Case 10/23, and 0.0028, 0.0029, 0.0017 sr$^{-1}$ for Case 10/25. Using the interpolated value of AERONET Rrs at RSP bands, the difference between RSP and AERONET Rrs are within 0.0003 sr$^{-1}$.
> …"

More discussion on comparison of SPEX Rrs and AERONET Rrs can be found in the revised file and the diff file.

The above criticisms notwithstanding, the study is interesting. The procedures for estimating the atmospheric interference are well defined. I would recommend showing retrievals over the entire 2 flights (along and perpendicular to the coast) to capture varied aerosol and water reflectance situations, even though in situ measurements may not be available, compare the remote sensing reflectance retrievals with those of the standard algorithm, and evaluate against the aircraft lidar measurements and satellite products, but this would require a new submission.

Thank you for the interest in the study and the suggestions to include the whole flight retrieval.

We have conducted studies on a flight track over water of day 10/23, and compared the retrieved RSP AOD (with polarization measurement) with the HSRL AOD as shown in the Figure 1 below. Over the flight track, the AOD variations are very small mostly around 0.02-0.04 (HSRL 532nm). For the day of 10/25, there are a limited number of pixels over water for analysis (plot not shown), and the AOD are around 0.03~0.05(HSRL 532nm). Therefore we are not discussing these results in the manuscript to capture aerosol variations, instead we only focus on the representative cases. For the study using standard atmospheric correction algorithm, as we have discussed previously, it requires new development of lookup table which is outside current study scope.

[Figure]

Figure 1. Retrieved RSP AOD and the HSRL AOD for Day 10/23. The green line indicates the location of the AERONET site where AERONET AOD coincides with the retrieved RSP AOD; grey area indicates the location of the island with data screened.

**Response to the report of reviewer 2:**

Thank the reviewer for the valuable suggestions and comments, which improve the clarity of this work significantly. On the technical correction/improvements, the figures and table have been updated and captions have been revised. On the scientific correction/improvements, the comments have been addressed below and the manuscript is revised accordingly.

This manuscript provides uses actual, airborne data sets obtained from the ACEPOL campaign to describe a proof-of-concept method for ocean color retrievals from the polarimeters and hyperspectral imagers onboard the 2022 NASA/PACE mission. The objectives, data sets, retrieval method, and validation efforts are explained clearly, and the subject of this manuscript falls well within the scope of AMT. Therefore, this manuscript is suitable for access review. The comments below provide suggestions and questions to improve the technical and scientific qualities of this manuscript

**Suggestions for technical corrections/improvements:**

1. **Fig 1b:** provide symbols for the polar angles that identify the numbers in this polar plot. Also, use a less-confusing definition for the asterisks (e.g., "back-scattering direction" or "sunglint" instead of "solar direction")

   Provided. The relative azimuth and zenith angles are indicated by $\phi$ and $\theta$ in the plot. Asterisk symbol indicates the antisolar point as revised in the figure caption.

2. **Fig. 4a**: mark the "(Chi_min)^2" values described in the text

   The minimum cost function values are indicated by arrows in both Fig 4a and 4b (for consistency).

   In order to align (Chi_min)^2 with the left edge of the first bin in the histogram, the histogram and cumulative probabilities are recomputed using bins from (Chi_min)^2 instead of zero as previously used.

3. **Table 3**: add "AOD(fine)" and "AOD(coarse)" for the optical thickness of fine-mode and coarse-mode aerosol, respectively
   Both fine and coarse mode AOD are added.

**Suggestions for scientific corrections/improvements:**

    1. **Page 2, Lines 15-16**:
   "… improve the retrieval performance of aerosol microphysical properties (Mischenko and Travis, 1997; …"
   ►
   "… improve the retrieval performance of aerosol microphysical properties (Mischenko and Travis, 1997; Chowdhary et al., 2001; …"
   Added. Thank you for suggesting the reference.

    2 Page 2, Line 24:
   "… and National Auronautics and Space Administration (NASA)'s Multi-Angle …"

[Figure]

"… the National Auronautics and Space Administration (NASA) Multi-Angle …

Updated.

**3 Page 3, Lines 11-14:**
"… To date, the procedures for using MAP data to aid the hyperspectral atmospheric correction of collocated ocean measurements … has not yet been demonstrated. …"
**Comment:**
This is not true: see https://ui.adsabs.harvard.edu/abs/2018AGUFMOS11D1435C/abstract. The lead authors of the current submitted manuscript may recall that they visited that AGU presentation and engaged in discussions. The citation is: Chowdhary, J., Stamnes, S., Zhang, M., Scarino, A.J., Wasilewski, A.P., Cairns, B., "Combining multispectral VIS-SWIR polarimetry and UV-NIR hyperspectral imagery to retrieve aerosol and ocean color properties from remote sensing: case studies for airborne RSP and GCAS observations", American Geophysical Union, Fall Meeting 2018, abstract #OS11D-1435

Thank you for the comments and reminding us the interesting work. We revised the manuscript as follows:

"…To date, there are only a few studies on performing atmospheric correction for hyperspectral radiometer using aerosol properties retrieved from the co-located MAP measurements. This is primarily due to the limited availability of co-located MAP and hyperspectral radiometer measurements over ocean. One such dataset is available from the North Atlantic Aerosols and Marine Ecosystems Study (NAAMES) field campaign in 2015, where both the GEO-CAPE Airborne Simulator (GCAS) (a hyperspectral radiometer) and RSP were deployed. These datasets have been used to study the hyperspectral ocean color retrievals (Chowdhary et al. 2018)."

**4 Page 4, Lines 8-9:**
"… advantageous in scenarios where the aerosol properties in the VIS or ultraviolet (UV) bands cannot be accurately extrapolated from measurements in the NIR-SWIR spectral range."

[Figure]

"… advantageous in scenarios where the aerosol properties in the VIS or ultraviolet (UV) bands cannot be accurately extrapolated from measurements in the NIR-SWIR spectral range (Chowdhary et al., 2019)."
Added. Thanks.

**5 Page 6, Line 20:**
"… depends on the polarization state and the water conditions (Zhai et al., 2017)."
►  _
"… depends on the wavelength and the water conditions (Chowdhary et al., 2012; Zhai et al., 2017)."
Citation:
Chowdhary, J., B. Cairns, F. Waquet, K. Knobelspiesse, M. Ottaviani, J. Redemann, L. Travis, and M. Mishchenko, 2012: Sensitivity of multiangle, multispectral polarimetric remote sensing over open oceans to water-leaving radiance: Analyses of RSP data acquired during the MILAGRO campaign. *Remote Sens. Environ.*, **118**, 284-308, doi:10.1016/j.rse.2011.11.003.
Added. Thanks.

6 **Page 7, Lines 8-9**:
"… where the waters are mostly clear so that the bio-optical model parameterized by Chlorophyll-a concentration is used. …"
**Comment:**
Clear waters do not always imply that their IOPs co-vary with the Chlorophyll- a concentration. For example, one of the clearest ocean waters are found off the West Coast of South America. These waters exhibit also a large deficiency in CDOM compared to other open ocean waters with the same Chlorophyll concentration (Morel, Claustre, Antoine, and Gentili, 2007; Morel, Gentili, Claustre, Babin, Bricaud, Ras, and Ti`eche, 2007). Given that the SPEXone-derived Rrs values so low in the blue spectrum when compared to MODIS and AERONET for Case 10/23 (see Fig. 6), might it be that this 1-parameter bio-optical model was not appropriate for this case study? In other words, did the authors also consider Rrs retrievals using their multi-parameter bio-optical model?

Thanks for the discussion. The multi-parameter bio-optical model proposed in our previous work (Gao et al, OE, 2018) was also investigated, but due to the small water leaving signals, there are large uncertainties associated with this complex model. A similar example has been demonstrated in Gao et al, AMT, 2019.  The Rrs from RSP with one-parameter bio-optical model discussed in this study agree well with the in-situ AERONET OC product. Note that the Rrs from SPEX use the same aerosol properties retrieved from RSP to conduct atmospheric correction.

7 **Page 7, Line 13**:
"… For a general study, Fu et al discussed …"
►

"… For a general study, Fu and Hasekamp discussed …"
Updated. Thanks.

8 **Page 10, Lines 20-21**:
"… which indicates the measurements cannot be modelled by the forward model …"
**Comment:**
One possibility may be a change of residual sunglint in nadir-viewing direction that is caused by wind-directionality of the ocean surface roughness. This is consistent with the reflectance results at 2250 in Figure 3(a) (which are predominantly sensitive to coarse-mode aerosols and sunglint contamination), and with the comments made in page 13, lines 10-11, on the sensitivity of sunglint to wind speed.
We agree with the reviewer.  Page 9, line 1-3 discussed that a scalar wind speed which represents an isotropic Cox-Munk model may be not sufficient to model the sunglint in this study. This is the reason why the sunglint data is not used in the current retrievals.

9 **Page 13, Line 5**:
"… SSA spectra for both fine and coarse modes …"
**Comment:**
Table 3 reports also the total SSA. How is this SSA value computed from those reported for the fine and coarse mode aerosols?
We added a sentence to define the total SSA following Bohren and Huffman(1998, page 445):

"The overall SSA for the two mode mixture is computed as the ratio of the number density weighted averages for the scattering and extinction cross sections (Bohren and Huffman, 1998). "

10 **Page 15, Line 8**:
"… which agree better with the AERONET results of 1.6. …"
*Comment:*
Aren't the AERONET retrievals assuming the same refractive index for the fine- and coarse-mode aerosols? Wouldn't this make the AERONET refractive index retrieval more sensitive to the dominant aerosol mode? To better assess the comparison of refractive index (and SSA!) retrievals with the AERONET products, Table 3 could therefore also report retrieval results AOD(fine) and AOD(coarse). This will allow one to compute a weighted refractive index and compare this with AERONET retrievals akin to Hasekamp et al. (2011).

Thanks for the suggestion. In order to be consistent with the reference of Hasekamp et al. (2011) and our previous study in Gao et al (2018), we computed the volume averaged refractive index and made the following revision in the manuscript:

"To compare with the AERONET refractive index which assumes both modes are the same, we define the volume-averaged refractive index as mv=fv * mr(fine)+(1-fv) *mr(Coarse) where fv is the fine mode volume fraction (Hasekamp et al 2011, Gao et al 2018). For Cases 10/23 and 10/25 with 7rho_t, mv is 1.49 and 1.48 respectively. While with 7rho_t+5P_t, mv becomes 1.58 and 1.56 for these two days, which agree better with the AERONET refractive index of 1.6."

11 **Page 16, Lines 19-20**:
"… The RSP and SPEX Rrs at wavelengths shorter than 470 are not compared due to the large absolute systematic difference in the 410 and 470 nm bands as discussed in Section 2. …"
*Comment:*
Why choosing to exclude the results for 410 nm? Since both the 410 and 470 nm bands show systematic differences, it makes more sense to either exclude, or include, results for *both* these bands.

Thank you for the question. We clarified this in the following revisions:

"The RSP 410nm band is excluded from comparison due to the observation of 4% decrease in its radiometric throughput, while other RSP bands maintain stable within ~ 1% in the radiometric calibration and ~ 0.1% in the polarimetric calibration (Knobelspiesse et al, The Aerosol Characterization from Polarimeter and Lidar (ACEPOL) airborne field campaign, to be submitted). The SPEX Rrs at wavelengths shorter than 470 nm are not compared because of the observed absolute systematic difference comparing with the RSP 410 and 470 nm bands as discussed in Section 2."

12 **Page 16, Line 22**:
"… Case 10/25 shows good agreement between SPEX Airborne, RSP, …"
*Comment:*

Specifically, this is also true for the 470 band. So why is this not true for the 470 band in Case 10/23, even after assigning an uncertainty bar for SPEX retrievals that is the same as the uncertainty bar for RSP retrievals in coinciding bands? Both these cases are only 2 days apart, which suggests that the absolute systematic difference between SPEX and RSP radiance measurements in the 470 band is the same for these days and can therefore not be the only cause for the disagreement in their Rrs retrievals.

Thank you for the comments. As the reviewer correctly pointed out that SPEX and RSP Rrs at 470nm agrees better in 10/25 than 10/23. Since the atmospheric correction for both RSP and SPEX use the same aerosol models retrieved from RSP, the difference in Rrs is more likely originated from the difference in their radiometric measurements. Note that the systematic and random difference between RSP and SPEX at 470nm band is 3% and 2% as shown in Table 2, Smit et al, 2019. We added the following discussion in the manuscript:

"Based on the definition of R_rs, the 4% and 3% systematic difference in the reflectance will transfer into Rrs biases around 0.002 and 0.0009 sr^{-1}. The random difference between RSP and SPEX measurements at 470nm band is 2% as discussed by (Smit et al 2019) which can transfer to 0.0006 sr-1 in Rrs. The differences of the R_rs} from RSP and SPEX at 470nm (0.0012 for Case 10/23 and 0.0003 for Case 10/25 with 7\rho_t+5P_t) may be due to the combined effects of the random and systematic differences in their measurements."

12 **Page 17, Lines 4-5**:
"… and the MODIS Rrs is in between …"
***Comment:***
Note that the MODIS and AERONET retrievals do not even agree with each other within their reported error bars. This suggests that the reported error bars for the MODIS and/or AERONET products are too small, which complicates a comparison with RSP/SPEX products. To assess one possible cause for MODIS-AERONET disagreements that is relevant to analyses of SPEX/RSP products, it might be useful to compare aerosol models used for atmospheric correction in MODIS-AERONET-RSP retrievals, and to discuss the impact of their differences on the Rrs retrievals shown in Fig. 6. Including in Fig. 6 the RSP and SPEX retrievals of Rrs for 410 nm would be useful for such a discussion even if RSP and SPEX radiances to not agree for this band (after all, one of them could still be correct).

Thank you for the comments.  The difference between SPEX and RSP Rrs have been discussed in the last comment. The reason why RSP 410nm is not used has been discussed in response to comment 11. The differences in their Rrs may due to the uncertainties in the measurement and also possible different aerosol models used for atmospheric correction. We revised the manuscript as follows:

*"The difference of RSP, SPEX and MODIS Rrs at wavelengths smaller than 500 nm may be related to the measurement uncertainties where the effects are larger for the same percentage uncertainties due to the larger total measurement values Another possible reason for the discrepancy between the MODIS Rrs and others is the different aerosol models used for atmospheric correction. For MODIS, it is determined from the two NIR bands of 748 and 869 nm while others are based on polarimeter retrievals."*

Note that the error bar for AERONET and MODIS are computed differently as discussed in section 2, which are generally larger than the reported uncertainties by including temporal and spatial variations. We revised section 2 to make it clearer as follows:

*"In order to evaluate the spatial variations when comparing with the retrieved water leaving reflectance, we averaged the MODIS (on board Aqua) water leaving reflectance within a 2km region around the USC_SEAPRISM site and compute its standard deviation as its maximum uncertainty. If smaller than 5%, the uncertainty is adopted as 5% which is the accuracy goal for blue band and clear water (Hu et al 2013). The AERONET measurements are available in almost every hour and there are a total of 8 measurements each day. The AERONET product provides good temporal coverage of the aerosol and ocean reflectance. We averaged the one-day AERONET products and compared its mean with the retrieval results, where the standard deviation (6% to 10% for both Cases) is used to represent the maximum uncertainties. Note that the reported uncertainty for AERONET OC Rrs is approximately 5% between 410- 550nm (Zibordi et al 2009)."*

13 **Page 17, Line 14**:
"… for highly spectrally resolved measurements of the ocean. …"
►

"… for hyper-spectrally resolved retrievals of the ocean color. …"
Thanks for the suggestion. We revised the sentence as:
"…perform an atmospheric correction **on** highly spectrally resolved measurements of the ocean."

14 **Page 18, Lines 25-26**:
"… The large uncertainties in coarse mode aerosol properties may be due to … the neglect of the SWIR DoLP in retrievals. …"
***Comment:***
This contradicts the statement given **Page 8, Lines 19-20**: "We have tested the effects of the DoLP at the two SWIR bands on the aerosol retrieval and found that including them does not improve the retrieval accuracies".
Thank you for this comment. As shown in Table 2, the total uncertainties in SWIR DOLP measurement are large with values around 6-15% (partly come from the average of five pixels together used in this study). Data with better SWIR DOLP accuracy should help coarse mode retrievals.

We modified the manuscript as follows to be more specific:
*"The large uncertainties in coarse model aerosol properties may be due to the small contribution of the coarse mode signal in the total reflectance and the large total uncertainties in the SWIR DoLP (not used in retrievals)."*

15 **Page 18, Lines 28-34; Page 19, Lines 5-6:**
"… This approach provides a statistical evaluation of the uncertainties relating to the cost function sensitivity and impact of initial values. … The study by Knobelspiesse et al. (2012) estimated retrieval uncertainties using the error propagation method for the Aerosol

Polarimetry Sensor (APS) … The uncertainties of the retrieval parameters evaluated using random initial values may not represent the full uncertainties in the retrieval. However, the two uncertainty results […] are comparable, which may relate to the intrinsic sensitivity of the cost functions when converged …"

**Comment:**

The authors are upfront and clear about how they compute error bars for the RSP products in this study. A thorough discussion on the merits of this method may fall beyond the scope of this manuscript. However, it is evident that such time-consuming computations, which require performing many retrievals for a single pixel, will not be practical for rapid inversions of PACE data. Rather, solving the cost function in Eq. (1) by means of Jacobians, and computing the error covariance matrix from these Jacobians, not only provides retrieval products but also their associated uncertainties. Given that this study was performed in context of the PACE mission, it might be appropriate that the authors provide a brief comment on this. Furthermore, Knobelspiesse et al (2012), who base their analyses on error covariance matrices, note that their error estimates represent the best possible retrieval uncertainties. Does this then not suggest that the retrieval uncertainties reported in this study are actually underestimated if they compare well with those reported in Knobelspiesse et al (2012)?

Thank you for the comment and suggestions. We have revised the manuscript with more discussions as follows:

[revised manuscript text omitted]